# Self-Reported Depressive Symptoms and Suicidality in Adolescents with Attention-Deficit/Hyperactivity Disorder: Roles of Bullying Involvement, Frustration Intolerance, and Hostility

**DOI:** 10.3390/ijerph18157829

**Published:** 2021-07-23

**Authors:** Tai-Ling Liu, Ray C. Hsiao, Wen-Jiun Chou, Cheng-Fang Yen

**Affiliations:** 1Department of Psychiatry, School of Medicine, College of Medicine, Kaohsiung Medical University, Kaohsiung 80708, Taiwan; tlliu@kmu.edu.tw; 2Department of Psychiatry, Kaohsiung Medical University Hospital, Kaohsiung 80708, Taiwan; 3Department of Psychiatry and Behavioral Sciences, University of Washington School of Medicine, Seattle, WA 98195-6560, USA; rhsiao@u.washington.edu; 4Department of Psychiatry, Children’s Hospital and Regional Medical Center, Seattle, WA 98105, USA; 5School of Medicine, Chang Gung University, Taoyuan 33302, Taiwan; 6Department of Child and Adolescent Psychiatry, Chang Gung Memorial Hospital, Kaohsiung Medical Center, Kaohsiung 83301, Taiwan

**Keywords:** adolescents, attention-deficit/hyperactivity disorder, cyberbullying, depression, frustration intolerance, hostility, suicidality, traditional bullying

## Abstract

This study examined the relationships of cyberbullying and traditional bullying victimization and perpetration, perceived family function, frustration discomfort, and hostility with self-reported depressive symptoms and suicidality in adolescents diagnosed as having attention-deficit/hyperactivity disorder (ADHD). Both the self-reported severity of depressive symptoms on the Center for Epidemiological Studies Depression Scale and the occurrence of suicidal ideation or a suicide attempt on the suicidality module of the Kiddie Schedule for Affective Disorders and Schizophrenia were assessed in 195 adolescents with ADHD. The adolescents completed the Cyberbullying Experiences Questionnaire, Chinese version of the School Bullying Experience Questionnaire, Frustration–Discomfort Scale, Buss–Durkee Hostility Inventory, and Family APGAR Index. Caregivers completed the ADHD problems component of the Child Behavior Checklist for Ages 6–18. Multiple regression analyses were used to examine the correlates for each of self-reported depressive symptoms and suicidality. The results showed that after the effects of gender, age, ADHD symptoms, and family function were controlled, greater frustration discomfort and bullying perpetration significantly predicted self-reported depressive symptoms. Being cyberbullying victims and displaying hostility significantly predicted the risk of suicidality. Various types of bullying involvement, frustration intolerance, and hostility significantly predicted self-reported depressive symptoms and suicidality in adolescents with ADHD. By monitoring and intervening in these factors, we can reduce the risk of depression-related problems and suicidality in adolescents with ADHD.

## 1. Introduction

### 1.1. Depressive Symptoms and Suicidality in Adolescents with Attention-Deficit/Hyperactivity Disorder

Attention-deficit/hyperactivity disorder (ADHD) is the most common neurodevelopmental disorder and is associated with adverse outcomes, including poor academic performance, mental and substance use disorders, criminality, and unemployment [1]. Depressive symptoms and suicidality are prevalent in individuals with ADHD. Longitudinal studies have demonstrated that children and adolescents with ADHD have a significantly higher risk of developing major depressive disorder compared with those without ADHD [2,3]. For example, girls with ADHD have a 2.5 times higher risk for major depressive disorder in adolescence and young adulthood compared with those without ADHD [2]. Children with ADHD at 4 to 6 years of age were at greatly increased risk for major depression or dysthymia (hazard ratio: 4.32) up to the age of 18 years relative to comparison children [3]. A US nationally representative household survey reported that adults with ADHD were five times more likely to develop a mood disorder than those without ADHD [4]. The 2007 National Survey of Children’s Health in the United States found that children and adolescents with ADHD were more likely to have depression than those without ADHD (14% vs. 1%) [5].

The significant relationship between ADHD and mood disorders was confirmed by a meta-analysis [6]. Prospective studies also confirmed that individuals with ADHD have a higher risk of suicide attempts than those without ADHD [3,7]. For example, children with ADHD at 4 to 6 years of age were at greatly increased risk for attempting suicide (hazard ratio: 3.60) up to the age of 18 years relative to comparison children [3]. Moreover, children and adolescents with concomitant ADHD and major depressive disorder have significantly greater impairment in their social and academic functioning [8], as well as more severe psychopathology and higher rates of long-term impairment than those with either disorder alone [9].

The lifetime prevalence of ADHD among children in Taiwan is 10.1% according to the Diagnostic and Statistical Manual of Mental Disorders, Fifth Edition (DSM-5), diagnostic criteria [9] in a nationally representative sample of children in Taiwan [10]. A nationwide population-based study in Taiwan demonstrated that individuals with ADHD showed a significantly increased probability of developing a depressive disorder when compared to the control group (ADHD: 5.3% vs. control: 0.7%) [11]. Another nationwide population-based study in Taiwan also showed that individuals with ADHD showed higher mortality caused by suicide (adjusted hazard ratio: 2.09; 95% confidence interval (CI): 1.62–2.71) than those without ADHD [12]. The results of previous studies supported these results, showing that depression-related problems and suicidality warrant the careful evaluation of individuals with ADHD.

### 1.2. Factors Related to Depression and Suicidality in Adolescents with ADHD

Determining the factors predicting depressive symptoms and suicidality in adolescents with ADHD is essential for developing effective prevention and intervention programs. Research has revealed that female sex [3], ADHD symptoms [13], genes [2,3], stressful life events [14], low self-competency [15], information-processing biases [14], maternal depression [3], parent–child difficulties [16,17], and problems with peers [16,17] can predict depression-related problems in individuals with ADHD. Studies have also shown that female sex [2,3], increased age [18], ADHD symptoms [19], poor family function [20], executive function deficits [21,22], comorbid depressive disorders, behavioral and substance use disorders [23,24], and maternal depression [3] can predict suicidal behaviors.

According to ecological systems theory [25], children develop within a complex system of relationships at various levels of their environment. The results of a previous study indicated that depressive symptoms and suicidality arise from interactions between children with ADHD and their social environments. Both individual (e.g., gender, age, ADHD symptoms, self-competency, and information-processing biases) and environmental factors (e.g., maternal depression, parent–child difficulties, and problems with peers) contributed to depressive symptoms and suicidality in individuals with ADHD. Therefore, depressive symptoms and suicidality in individuals with ADHD should be considered for prevention and intervention from an ecological viewpoint. In addition to the individual and environmental factors that have been examined in previous studies, the roles of victimization and perpetration of cyberbullying and traditional bullying, frustration intolerance, and hostility in depressive symptoms and suicidality in adolescents with ADHD warrant further study.

### 1.3. Relationships of Cyberbullying and Traditional Bullying Victimization and Perpetration with Depressive Symptoms and Suicidality in Adolescents with ADHD

Bullying is the activity of repeated, aggressive behavior that is intended to physically, mentally, or emotionally hurt another individual [26]. Traditional bullying can involve physical acts, verbal utterances, social exclusion, property theft, and other behaviors [27]. A meta-analysis provided strong evidence for a causal relationship of traditional bullying victimization with depressive symptoms and suicidal ideation and suicide attempts [28]. A previous study on 6406 adolescents in Taiwan also confirmed the positive association between victimization and perpetration of traditional bullying with self-reported depressive symptoms [29]. Cyberbullying refers to bullying behaviors that are perpetrated through electronic means [30]. Although another meta-analysis revealed a positive and significant relationship between depressive symptoms and cyberbullying victimization [31], their temporal relationships are currently unclear. A few longitudinal studies have indicated that cyberbullying victimization predicts depressive symptoms [32,33], whereas others have suggested that depressive symptoms predict cyberbullying victimization [34,35]. One study confirmed that cyberbullying victimization and depressive symptoms had a reciprocal relationship [36]. Comparatively, the roles of cyberbullying perpetration and victimization in suicidality have received less attention. A recent study revealed that although both victimization and perpetration of traditional bullying and cyberbullying were cross-sectionally associated with suicidal ideation and suicide attempts, only perpetration of traditional bullying and cyberbullying prospectively predicted suicidal ideation or attempts 1 year later [37].

Limited studies have examined the relationships of cyberbullying and traditional bullying involvement with depressive symptoms and suicidality in adolescents with ADHD. A study of adolescents with ADHD reported that victimization via traditional bullying was significantly associated with depressive symptoms [13]. Similarly, research on adolescents with ADHD showed that the perpetration of traditional bullying was significantly associated with depressive symptoms [13] and suicidal ideation [18]. A study of male adolescents with ADHD revealed that cyberbullying victims reported more severe depressive symptoms and suicidality than those who were not cyberbullying victims [38]. However, no study has simultaneously examined the roles of cyberbullying and traditional bullying involvement in depressive symptoms and suicidality in adolescents with ADHD.

### 1.4. Relationships of Frustration Intolerance and Hostility with Depression and Suicidality in Adolescents with ADHD

Frustration intolerance is a type of irrational belief [39]. People with high frustration intolerance find it difficult to accept that reality does not correspond to personal desires [40]. For example, people with high frustration intolerance may strongly demand that existing conditions must be changed to give them what they like; otherwise, they cannot stand it at all [41]. A 6.5-year follow-up study reported that frustration intolerance positively predicted the severity of depressive symptoms [42]. Research also revealed that frustration intolerance is a crucial personality trait in adolescents with suicidal behavior [43]. Hostility denotes an emotional and expressive characteristic that indicates the potential intent to be aggressive toward and assault others [44]. Research has shown that hostility is associated with increased risks of depressive disorders [45] and suicidality [46]. Individuals with ADHD have higher frustration intolerance [47,48,49] and hostility [50,51] than those without ADHD. The relationships of frustration intolerance and hostility with depressive symptoms and suicidality in adolescents with ADHD, however, remain uninvestigated.

### 1.5. Study Aims

This study examined the associations of cyberbullying and traditional bullying victimization and perpetration, frustration discomfort, and hostility (independent variables) with self-reported depressive symptoms and suicidality (dependent variables) in adolescents with ADHD by controlling for the effects of gender, age, ADHD symptoms, and perceived family function (covariates). Accordingly, the specific hypotheses were as follows:

**Hypothesis** **1 (H1).**
*Adolescents with ADHD who*
*are victims or perpetrators of*
*cyberbullying and traditional bullying have more severe*
*self-reported depressive symptoms and are more likely to report suicidality than nonvictims and nonperpetrators.*


**Hypothesis** **2 (H2).**
*Greater frustration discomfort and hostility significantly predict*
*self-reported depressive symptoms and suicidality in adolescents with ADHD.*


## 2. Methods

### 2.1. Participants

The participants were recruited from two outpatient clinics in the child psychiatric departments of two hospitals in Kaohsiung, Taiwan, between June 2019 and January 2021. The inclusion criteria were (1) aged 11–18 years and (2) diagnosis of ADHD according to the diagnostic criteria specified in DSM-5 [9] and based on diagnostic interviews conducted by child psychiatrists. Caregivers of the chosen adolescents were also included in this study. Adolescents and caregivers who had schizophrenia, bipolar disorder, intellectual disabilities, autism spectrum disorder, cognitive deficits, or communication difficulties that adversely affected their ability to understand the study purpose or complete the questionnaires were excluded. In total, 216 adolescents with ADHD and their caregivers visited the outpatient clinics during the period of study. Of them, 8 adolescents were excluded based on the exclusion criteria. A total of 208 adolescents with ADHD and their caregivers were consecutively recruited, and 195 (93.8%) consented to participate. This study was approved by the Institutional Review Boards of Kaohsiung Chang Gung Memorial Hospital (approval number: 201900432A3; date of approval: 3 June 2019) and Kaohsiung Medical University Hospital (approval number: KMUHIRB-SV(I)-20190034; date of approval: 17 May 2019). All adolescents and caregivers provided written informed consent before enrollment.

### 2.2. Measures

#### 2.2.1. Dependent Variables

##### Self-Reported Depressive Symptoms

The 20-item Mandarin Chinese version of the Center for Epidemiological Studies Depression Scale (CES-D) was used to assess the severity of self-reported depressive symptoms, including depressed/negative affect (e.g., “I felt depressed.”), positive affect (reverse scoring, e.g., “I was happy.”), somatic and retarded activities (e.g., “I had trouble keeping my mind on what I was doing.”), and interpersonally negative relations (e.g., “I felt that people disliked me.”) [52,53]. Adolescents were asked how often they experienced each symptom in the preceding month. The response categories were 0—rarely or none of the time (less than 1 day), 1—some or a little of the time (1–2 days), 2—occasionally or a moderate amount of the time (3–4 days), or 3—most or all of the time (5–7 days). The total score indicates the severity of self-reported depressive symptoms. The Mandarin Chinese version of the CES-D has good validity regarding discriminating both major depressive disorder (area under the ROC curves (AUCs) = 0.90) and dysthymic disorder (AUC = 0.94), excellent internal consistency (α = 0.93), and a good test–retest reliability (*r* = 0.78) among Taiwanese adolescents in the community [54,55,56]. In the current sample, the internal consistency was good (α = 0.84).

##### Suicidality

The five items constituting the suicidality module of the Kiddie Schedule for Affective Disorders and Schizophrenia [57] were used to assess suicidal ideation (e.g., “Has there ever been a period of 2 weeks or longer when you thought a lot about death, including thoughts of your own death, somebody else’s death, or death in general?”) and suicide attempts (e.g., “Have you ever attempted suicide?”) in the preceding year [58]. A previous study on Taiwanese adolescents reported that its validity was acceptable (kappa coefficient of agreement between adolescents’ self-reports and their parents’ reports: 0.541; *p* < 0.001) [58]. Each question elicited a “yes” or “no” response. Participants responding ‘‘yes’’ to any of the five items were classified as having suicidality.

#### 2.2.2. Independent Variables

##### Cyberbullying Victimization and Perpetration

The six-item Cyberbullying Experiences Questionnaire (CEQ) was used to assess adolescents’ self-reported experiences of perpetrating or being victimized by the posting of mean or hurtful comments, upsetting pictures, photos, or videos, and the spreading of rumors on social media or through emails, images, video clips, or blogs in the previous year [38]. Each item was rated using a 4-point Likert-type scale ranging from 0 (*never*) to 3 (*all the time*). Because of the skewness of the data, participants who scored 1 or above to any of the first three or final three items were identified as cyberbullying victims or perpetrators, respectively. A previous study on adolescents with ADHD in Taiwan using the cutoff of the CEQ demonstrated the significant association between cyberbullying victimization and traditional bullying victimization, as well as the associations of cyberbullying perpetration with traditional bullying perpetration and internet addiction (α = 0.70 for victimization and 0.64 for perpetration) [38]. In the current sample, the internal consistencies (α’s) were 0.70 and 0.65 for cyberbullying victimization and perpetration, respectively.

##### Traditional Bullying Victimization and Perpetration

The 16-item Mandarin Chinese version of the School Bullying Experience Questionnaire (C-SBEQ) was used to assess the adolescents’ self-reported experiences of traditional bullying victimization and perpetration in the previous year [59]. The first eight items assessed experiencing social exclusion, offensive name-calling, ill-speaking, physical abuse, forced work, and the confiscation of money, daily supplies, and snacks (e.g., “How often have others spoken ill of you?” “How often have others beaten you up?”). The final eight items addressed experiences of perpetration that were mentioned in the first eight items. Each item was scored on a 4-point Likert scale ranging from 0 (*never*) to 3 (*all the time*). Because of the skewness of the data, participants who scored 2 or 3 points for any of the first eight or final eight items were identified as victims or perpetrators, respectively, of social, verbal, or physical bullying. The C-SBEQ has acceptable validity regarding discriminating victims and perpetration (kappa coefficient of agreement between adolescents’ self-report and their teachers’ and classmates’ nomination: 0.52 for victims and 0.45 for perpetrators; *p* < 0.001), acceptable internal consistency (α = 0.73 for victimization and 0.76 for perpetration), and acceptable test–retest reliability (*r* = 0.80 for victimization and 0.76 for perpetration) among adolescents in Taiwan [59]. In the current sample, the internal consistency was acceptable (α = 0.76 for victimization and 0.72 for perpetration).

##### Frustration Intolerance

The 28-item Mandarin Chinese version of the Frustration Discomfort Scale (FDS) was used to evaluate the adolescents’ self-reported frustration intolerance, including emotional intolerance, demands for entitlement, comfort, and achievement (e.g., “I can’t stand having to wait for things I would like now.”) [60,61,62]. Each item was rated on a 5-point Likert scale ranging from 1 (*absent*) to 5 (*very strong*). The total score indicates the level of frustration intolerance. The Mandarin Chinese version of the FDS has excellent internal consistency among adolescents within Taiwan (α = 0.90) [62]. In the current sample, the internal consistency was good (α = 0.88).

##### Hostility

The 20-item Buss–Durkee Hostility Inventory–Chinese Version–Short Form (BDHIC-SF) was used to assess the adolescents’ self-reported hostility cognition, hostility affect, expressive hostility behavior, and suppressive hostility behavior (e.g., ‘‘If somebody hits me, I hit back.’’) [63,64]. Each item was rated on a 5-point Likert-type scale ranging from 1 (*strongly disagree*) to 5 (*strongly agree*). The total score represents the level of hostility. The BDHIC-SF has excellent internal consistency (α = 0.93) and good test–retest reliability (*r* = 0.80) in the Taiwanese population [64]. In the current sample, the internal consistency was good (α = 0.85).

#### 2.2.3. Covariates

##### Demographic Characteristics

Adolescents’ gender (girls vs. boys) and age were collected.

##### ADHD symptoms

The caregiver-reported Chinese version of the Child Behavior Checklist (CBCL) for Ages 6–18 was used to measure the adolescents’ behavioral problems [65,66,67]. We used the recommended T-score transformations of the raw behavior scores, which adjust for age and sex differences in behavior found in normative samples. The ADHD symptom domains were used for the analysis. The CBCL for Ages 6–18 has good internal consistency (α = 0.82) in Taiwanese children and adolescents [67]. In the current sample, the internal consistency was good (α = 0.81).

##### Perceived Family Function

The five-item Mandarin Chinese version of the Family APGAR Index was used to assess adolescents’ perceived family support, including the components of adaptability, partnership, growth, affection, and resolve (e.g., “I am satisfied with the help that I receive from my family when something is troubling me.”) [68,69]. The total score represents the level of family support. The Mandarin Chinese version of the Family APGAR Index has excellent internal consistency (α = 0.88) in Taiwanese adolescents [68]. In the current sample, the internal consistency was good (α = 0.81).

### 2.3. Statistical Analysis

We performed the data analysis using IBM SPSS Statistics version 24.0 software (IBM Corp., Armonk, NY, USA). Adolescents’ gender, experiences of bullying involvement, and suicidality were expressed as percentages. Adolescents’ age, ADHD symptoms, perceived family function, frustration intolerance, hostility, and self-reported depressive symptoms were expressed as means and standard deviations (SDs). The relationships of experiences of bullying involvement, frustration intolerance, and hostility (independent variables) with self-reported depressive symptoms and suicidality (dependent variables) were examined using multiple regressions by using gender, age, ADHD symptoms, and perceived family function as covariates. R-squared (R^2^) was used to represent the effect size of the variables in the multiple regression analyses [70]. Odds ratios (ORs) and 95% CIs were used to represent statistical significance and effect sizes of the variables in the logistic regression analysis [71]. A two-tailed *p*-value of <0.05 indicated statistical significance.

## 3. Results

Table 1 presents the demographic characteristics, ADHD symptoms, social interaction, behavioral characteristics, self-reported depressive symptoms, and suicidality in 195 adolescents (31 girls and 164 boys; M_age_ = 13.5 years, SD = 2.3 years). The mean score of the self-reported depressive symptoms in the CES-D was 15.1 (SD = 9.6, range = 0–49); 26.7% of participants reported either suicidal ideation or an attempt in the preceding year. We used the Shapiro–Wilk test to examine the normality of depressive symptoms, frustration intolerance, and hostility. All *p*-values > 0.05, indicating they were normally distributed. Victims of traditional bullying were more likely to be victims of cyberbullying than non-victims of traditional bullying (χ^2^ = 9.275, *p* = 0.002), whereas no difference in the risk of being cyberbullying perpetrators was found between perpetrators and nonperpetrators of traditional bullying (χ^2^ = 1.346, *p* = 0.246).

The results of the multiple regression analyses examining the factors related to self-reported depressive symptoms are shown in Table 2. Model 1 demonstrated the relationships between covariates, including gender, age, ADHD symptoms, and perceived family function and the self-reported depressive symptoms (F (*df*) = 5.101 (4, 190), *p* = 0.001, change in R^2^ = 0.097). Model 2 included experiences of bullying involvement in addition to the covariates (F (*df*) = 6.372 (8, 186), *p* < 0.001, change in R^2^ = 0.118). The results of model 2 demonstrated that after controlling for the effects of the covariates, being perpetrators of traditional bullying significantly predicted self-reported depressive symptoms. Model 3 included frustration discomfort and hostility, in addition to the covariates and experiences of bullying involvement (F (*df*) = 10.772 (10, 184), *p* < 0.001, change in R^2^ = 0.154). The results of model 3 showed that frustration intolerance significantly predicted self-reported depressive symptoms. The condition index was 21.7, indicating that there was no multicollinearity problem.

The results of the logistic regression analysis examining the factors related to suicidality are shown in Table 3. Model 1 demonstrated the relationships between the covariates and suicidality. Model 2 included experiences of bullying involvement in addition to the covariates. The results of model 2 demonstrated that after controlling for the effects of the covariates, being cyberbullying victims significantly predicted suicidality. Model 3 included frustration discomfort and hostility, in addition to the covariates and experiences of bullying involvement. The results of model 3 showed that hostility significantly predicted the risk of suicidality.

## 4. Discussion

This study showed that traditional bullying perpetration and frustration intolerance significantly predicted self-reported depressive symptoms; in addition, cyberbullying victimization and hostility significantly predicted the risk of suicidality in adolescents with ADHD.

### 4.1. Relationships of Various Types of Bullying Involvement with Depressive Symptoms and Suicidality

Stressful life events can predict depressive symptoms in individuals with ADHD [14]. Because cyberbullying and traditional bullying victimization are stressful life events that commonly occur among adolescents, we hypothesized that ADHD adolescents who were victims of cyberbullying and traditional bullying would have more severe self-reported depressive symptoms and a higher risk of suicidality than nonvictims. However, we observed that suicidality was significantly predicted by cyberbullying victimization but not traditional bullying. A previous study similarly reported a significant association between suicidality and cyberbullying victimization—but not perpetration—in adolescents with ADHD [38]. Adolescents with ADHD may spend more time on the internet than those without ADHD because of several biopsychosocial mechanisms, including being easily bored, an aversion for delayed reward, frustration, poor interpersonal relationships in real-life situations, impaired inhibition, and a motivation deficit [72]; therefore, the risk of experiencing cyberbullying victimization is higher in adolescents with ADHD than in those without. Cyberbullying may negatively affect the self-esteem of victims with ADHD and compromise their mental health. Cyberbullying may also reduce the opportunities of these victims to make friends online and seek the social support required to prevent suicidality. Because the internet has become a vital living environment for adolescents in modern times, health professionals should routinely screen adolescents for cyberbullying victimization to promptly detect their risk of suicidality.

We observed that adolescents with ADHD who perpetrated traditional bullying had more severe self-reported depressive symptoms than nonperpetrators. A previous study similarly revealed that traditional bullying perpetrators who had ADHD were more likely to have suicidal intent than nonperpetrators [18], but another study of adolescents in a community setting indicated that both victims and perpetrators of traditional bullying had more severe depressive symptoms and a higher risk of suicidality than nonvictims and nonperpetrators [29]. Several possible explanations may account for these results. First, traditional bullying perpetration may worsen the social isolation of adolescents with ADHD, and social isolation is a significant predictor of suicide risk in adolescents [73]. Second, bullying perpetration may serve various social functions. Depending on these functions, perpetrators differ in their skills and mental statuses [74]. Social skill deficits [75], anger [76], and impulsivity [77] are common in adolescents with ADHD. Traditional bullying perpetration may be a product of poor social skills, anger, and impulsivity, which altogether may increase the difficulties of daily living for adolescents with ADHD and compromise their emotional regulation [78]. The results of this study indicate that mental health and education professionals must monitor mental health problems among not only ADHD adolescents who are victims of bullying but also those who perpetrate it.

### 4.2. Relationships of Frustration Intolerance and Hostility with Depression and Suicidality

This study showed that frustration intolerance and hostility significantly predicted self-reported depressive symptoms and suicidality, respectively, in adolescents with ADHD. Frustration intolerance is a type of irrational belief [39]. According to rational emotive behavior therapy [79], people who think less irrationally respond to daily stressors or hassles differently than do people who think more irrationally. Compared with adolescents without ADHD, adolescents with ADHD are more likely to have peer relationship problems [80,81] and poor academic performance [82]. Frustration intolerance may compromise the ability of adolescents with ADHD to develop rational coping strategies to manage stressors, thereby worsening emotional regulation. Our results indicate that ADHD adolescents with high frustration intolerance warrant interventions to reduce their risk of depression. Rational emotive behavior therapy was shown to be effective at alleviating frustration intolerance [83].

Similar to the results of a previous study conducted in a community setting [46], our study revealed that hostility significantly predicted suicidality in adolescents with ADHD. Several hypotheses may explain the result. First, hostility may worsen relationships with others for adolescents with ADHD, which further exacerbates interpersonal conflicts and aggravates suicidal intent. Second, hostility may prevent or delay adolescents with ADHD and suicidal intent from seeking support from their peers, families, and teachers. Third, adolescents who are highly hostile toward others may also have a hostile attitude toward themselves; suicide may serve as a hostile act directed inward to the self. Although our result highlights the necessity of interventions for hostility in adolescents with ADHD, the results of some previous studies did not confirm the effects of pharmacological treatment for ADHD on reducing the level of hostility in individuals with ADHD [84,85,86]. Psychological and pharmacological treatments targeting hostility’s adverse affective and behavioral effects on adolescents with ADHD remain challenges that warrant further investigation.

### 4.3. Limitations

This is one of the first studies to simultaneously examine the relationships of traditional bullying and cyberbullying victimization and perpetration, frustration intolerance, and hostility with self-reported depressive symptoms and suicidality in adolescents with ADHD. However, our study had several limitations. First, the cross-sectional research design limited our ability to draw conclusions regarding the causal relationships of bullying involvement, frustration intolerance, and hostility with depressive symptoms and suicidality. Second, the adolescents themselves provided the data for bullying involvement, frustration intolerance, hostility, depressive symptoms, and suicidality. The problem of shared-method variance occurs due to having a sole information source and requires careful consideration. Further studies are required to determine whether the associations change when other sources of information are used. Third, the results of this study may not be generalizable to adolescents who have not visited psychiatric units. Moreover, this study did not include adolescents without ADHD; therefore, we could not determine whether the associations of bullying involvement, frustration intolerance, and hostility with depressive symptoms and suicidality found in this study also exist in adolescents without ADHD. Fourth, although the CEQ used for measuring the involvement in cyberbullying in this study was used in previous studies on various groups, such as adolescents with ADHD [38], autism spectrum disorder [87], and gay and bisexual men [88], and could differentiate those with and without severe mental health problems related to experiences of cyberbullying involvement, its small number of items (three for victimization and three for perpetration) and borderline acceptable level of internal reliability (Cronbach’s alpha values 0.70 for victimization and 0.64 for perpetration) might limit its value of use in assessing experiences of cyberbullying involvement.

## 5. Conclusions

This study showed that various types of bullying involvement, frustration intolerance, and hostility significantly predicted self-reported depressive symptoms and suicidality in adolescents with ADHD. Both social and individual factors predicted self-reported depressive symptoms and suicidality. Mental health professionals should therefore consider depressive symptoms and suicidality in this group as products of individual–environment interactions. These social and individual factors could thus be integrated into the prevention and intervention strategies for reducing the risks of depression-related problems and suicidality in adolescents with ADHD.

## Figures and Tables

**Table 1 ijerph-18-07829-t001:** Dependent variables, independent variables, and covariates (*N* = 195).

Characteristics	*n* (%)	Mean (SD)	Range
Dependent variables			
Self-reported depressive symptoms on the CESD		15.1 (9.6)	0–49
Suicidal idea or attempt	52 (26.7)		
Independent variables			
Cyberbullying			
Victims	28 (14.4)		
Perpetrators	17 (8.7)		
Traditional bullying			
Victims	54 (27.7)		
Perpetrators	35 (17.9)		
Frustration discomfort on the FDS		68.4 (24.1)	28–129
Hostility on the BDSI-CS		55.2 (16.4)	20–94
Covariates			
Gender			
Girls	31 (15.9)		
Boys	164 (84.1)		
Age (years)		13.5 (2.3)	11–18
ADHD problems on the CBCL		61.7 (7.6)	40–80
Perceived family function		13.5 (4.2)	5–20

ADHD: attention-deficit/hyperactivity disorder; BDHIC-SF: Buss–Durkee Hostility Inventory–Chinese Version–Short Form; CBCL: Child Behavior Checklist; CESD: Center for Epidemiological Studies Depression scale; FDS: Frustration Discomfort Scale; SD: standard deviation.

**Table 2 ijerph-18-07829-t002:** Factors related to self-reported depressive symptoms: multiple regression analyses (*N* = 195).

Variables	Model 1		Model 2		Model 3	
	B (SE)	*p*	B (SE)	*p*	B (SE)	*p*
Gender	−3.273 (1.823)	0.074	−4.257 (1.732)	0.015 *	−3.840 (1.584)	0.016 *
Age	0.917 (0.295)	0.002 **	0.992 (0.284)	0.001 **	0.919 (0.258)	<0.001 ***
ADHD problems	0.104 (0.089)	0.244	0.041 (0.085)	0.626	0.057 (0.077)	0.455
Perceived family function	−0.349 (0.161)	0.032 *	−0.259 (0.153)	0.092	−0.238 (0.139)	0.087
Cyberbullying victims			3.514 (2.074)	0.092	1.974 (1.884)	0.296
Cyberbullying perpetrators			2.271 (2.569)	0.378	0.664 (2.329)	0.776
Traditional bullying victims			2.359 (1.591)	0.140	1.846 (1.443)	0.202
Traditional bullying perpetrators			5.328 (1.846)	0.004 **	3.467 (1.698)	0.043 *
Frustration discomfort					0.139 (0.033)	<0.001 ***
Hostility					0.051 (0.049)	0.299
F (*df*)	5.101 (4, 190)		6.372 (8, 186)		10.772 (10, 184)	
*p*	0.001 **		<0.001 ***		<0.001 ***	
Adjusted R^2^	0.078		0.181		0.335	
Change of R^2^	0.097		0.118		0.154	

SE: standard error. * *p* < 0.05, ** *p* < 0.01, *** *p* < 0.001.

**Table 3 ijerph-18-07829-t003:** Factors related to suicidality: logistic regression analysis.

Variables	Model 1		Model 2		Model 3	
	OR (95% CI)	*p*	OR (95% CI)	*p*	OR (95% CI)	*p*
Gender	0.693 (0.297–1.616)	0.396	0.553 (0.228–1.339)	0.189	0.607 (0.239–1.543)	0.294
Age	0.999 (0.866–1.153)	0.992	0.991 (0.850–1.156)	0.907	0.986 (0.837–1.161)	0.864
ADHD problems	0.995 (0.952–1.039)	0.815	0.982 (0.938–1.029)	0.454	0.982 (0.936–1.031)	0.467
Perceived family function	0.948 (0.877–1.025)	0.181	0.957 (0.882–1.039)	0.296	0.957 (0.877–1.045)	0.329
Cyberbullying victims			2.678 (1.011–7.093)	0.048 *	2.302 (0.848–6.252)	0.102
Cyberbullying perpetrators			2.711 (0.811–9.065)	0.105	2.216 (0.648–7.578)	0.204
Traditional bullying victims			1.475 (0.655–3.323)	0.348	1.300 (0.554–3.049)	0.547
Traditional bullying perpetrators			1.164 (0.457–2.968)	0.750	0.836 (0.312–2.241)	0.722
Frustration discomfort					1.009 (0.989–1.030)	0.361
Hostility					1.031 (1.001–1.064)	0.049 *

CI: confidence interval; OR: odds ratio. * *p* < 0.05.

## Data Availability

The data will be available upon reasonable request to the corresponding authors.

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
