# Peer review of "Self-Reported Depressive Symptoms and Suicidality in Adolescents with Attention-Deficit/Hyperactivity Disorder: Roles of Bullying Involvement, Frustration Intolerance, and Hostility"

_ijerph, 2021, doi:10.3390/ijerph18157829_

Round 1
Reviewer 1 Report
“Depression and Suicidality in Adolescents with Attention-Deficit/Hyperactivity Disorder: Roles of Bullying Involvement, Frustration Intolerance, and Hostility”
This is a really interesting and relevant scientific contribution, with important implications for the improvement of the quality of life on adolescents with ADHD.
Please provide with the following changes in order to improve the overall quality of the paper:
- Measures: please provide further details for all measures. Examples of items, operational definitions of the constructs measured, etc.
- Tables 2 and 4: please include (*/**) if p <0.05 or <0.001 respectively.
- Table 3: please include ANOVA statistics and level of significance.
Best wishes.
Author Response
We appreciated your valuable comments. As discussed below, we have revised our manuscript with underlines based on your suggestions. Please let us know if we need to provide anything else regarding this revision.
Comment 1
Measures: please provide further details for all measures. Examples of items, operational definitions of the constructs measured, etc.
Response
Thank you for your suggestion. We added examples of items and operational definitions of the constructs measured as below.
Depression: “depressive symptoms including depressed/negative affect (e.g., “I felt depressed.”), positive affect (reverse scoring, e.g., “I was happy.”), somatic and retarded activities (e.g., “I had trouble keeping my mind on what I was doing.”), and interpersonally negative relations (e.g., “I felt that people disliked me.”) Please refer to line 182-186.
Suicidal ideation and attempt: “e.g., “Has there ever been a period of 2 weeks or longer when you thought a lot about death, including thoughts of your own death, somebody else's death, or death in general?” “Have you ever attempted suicide?” Please refer to line 198-201.
Bullying victimization: “experiencing social exclusion, offensive name-calling, and ill-speaking, physical abuse, forced work, and confiscation of money, daily supplies, and snacks (e.g., “How often have others spoken ill of you?” “How often have others beaten you up?”). Please refer to line 225-227.
Frustration intolerance: “emotional intolerance, demands for entitlement, comfort, and achievement (e.g., “I can’t stand having to wait for things I would like now.”)” Please refer to line 241-243.
Hostility: “hostility cognition, hostility affect, expressive hostility behavior, and suppressive hostility behavior (e.g., ‘‘If somebody hits me, I hit back’’)” Please refer to line 250-252.
Family support: “perceived family support including the components of adaptability, partnership, growth, affection, and resolve (e.g., “I am satisfied with the help that I receive from my family when something is troubling me.”)” Please refer to line 270-272.
Comment 2
Tables 2 and 4: please include (*/**) if p <0.05 or <0.001 respectively.
Response
Thank you for your suggestion. We added the */**/*** to indicate the p value of < .05, < .01, and < .001, respectively into Tables. Please refer to line 316 and 326.
Comment 3
Table 3: please include ANOVA statistics and level of significance.
Response
In the revised manuscript we deleted the preliminary t-tests and Pearson’s correlation. Instead, we used a multiple regression for each dependent variable and obtained the similar results. Therefore, the original Table 3 was deleted.
Reviewer 2 Report
The study focuses on a relevant and timely topic, i.e. the relationships between bullying/cyberbullying participant roles and suicidality and depression in adolescents with ADHD. The manuscript is well written and clearly organized, but I have some mainly methodological concerns that I think should be addressed before it can be considered for publishing.
-
One of the most important results of this study concerns the relationship of cyberbullying victimization and suicidality in ADHD adolescents. However, the scale used for measuring the involvement in cyberbullying is a six items scale with three items for cyberbullying and three for cybervictimization. The Cronbach's alpha values reported are borderline acceptable and the cited study ("A previous study reported the reliability and validity of the CEQ as acceptable [27]") was authored by the same research group and is not a validation study, besides also reporting suboptimal Cronbach's alpha values for the two 3-items scales. In light of these facts, I don't think this result can be presented as the central finding of the study.
-
It is not clear whether a stepwise regression was actually performed (and, if so, why), especially as only variables already found to be associated with the DV were entered in this regression. I also don't understand the need for the preliminary t-tests. I think a multiple regression for each DV would probably provide similar results in a simpler and clearer fashion.
-
Please specify the softwares/packages used for the statistical analysis
-
In the abstract, the phrase "multiple and logistic regression analyses" is used. The analysis methods adopted are not clear: logistic regressions are also multiple regressions in this case. I suggest simply using "multiple regressions".
-
I suggest avoiding the phrasing "significant variables": it's not variables that can be significant, rather the relationships between variables
-
I wonder why no normative group was included in the study. It would have been useful to check whether the association between bullying-related variables and suicidality and depression are different (e.g. stronger) in ADHD individuals. In particular, authors seem interested in the comparison between ADHD and non-ADHD adolescents (see e.g. paragraph 4.1), so this would be very useful to test their hypotheses. I think this should be at least included as a limitation.
-
Scoring is not clear for some variables: e.g. Frustration intolerance: were means/medians used? The rationale for coding some variables binomially should also be specified.
Author Response
We appreciated your valuable comments. As discussed below, we have revised our manuscript with underlines based on your suggestions. Please let us know if we need to provide anything else regarding this revision.
Comment 1
One of the most important results of this study concerns the relationship of cyberbullying victimization and suicidality in ADHD adolescents. However, the scale used for measuring the involvement in cyberbullying is a six items scale with three items for cyberbullying and three for cybervictimization. The Cronbach's alpha values reported are borderline acceptable and the cited study ("A previous study reported the reliability and validity of the CEQ as acceptable [27]") was authored by the same research group and is not a validation study, besides also reporting suboptimal Cronbach's alpha values for the two 3-items scales. In light of these facts, I don't think this result can be presented as the central finding of the study.
Response
Thank you for your comment. We agree that although the Cyberbullying Experiences Questionnaire (CEQ) has been used to assess the experiences of cyberbullying involvement in various groups, its small number of items and borderline acceptable level of internal reliability might limit its value of use. We added more introduction for the CEQ and listed its small number of items and borderline acceptable level of internal reliability as one of limitations in this study as below.
“A previous study on adolescents with ADHD in Taiwan using the cutoff of the CEQ demonstrated the significant association between cyberbullying victimization and traditional bullying victimization as well as the associations of cyberbullying perpetration with traditional bullying perpetration and internet addiction (α = .70 for victimization and .64 for perpetration) [38].” Please refer to line 215-219.
“Fourth, although the CEQ used for measuring the involvement in cyberbullying in this study has been used in previous studies on various groups such as adolescents with ADHD [38], autism spectrum disorder [88], and gay and bisexual men [89] and could differentiate those with and without severe mental health problems related to experiences of cyberbullying involvement, its small number of items (three for victimization and three for perpetration) and borderline acceptable level of internal reliability (Cronbach's alpha values .70 for victimization and .64 for perpetration) might limit its value of use in assessing experiences of cyberbullying involvement.” Please refer to line 413-420.
Comment 2
It is not clear whether a stepwise regression was actually performed (and, if so, why), especially as only variables already found to be associated with the DV were entered in this regression. I also don't understand the need for the preliminary t-tests. I think a multiple regression for each DV would probably provide similar results in a simpler and clearer fashion.
Response
Thank you for your suggestion. In the revised manuscript we deleted the preliminary t-tests and Pearson’s correlation. Instead, we used a multiple regression for each dependent variable and obtained the similar results. We replaced the original tables 2 to 5 by tables 2 and 3 presenting the results of multiple regressions for depressive symptoms and suicidality, respectively. We also rewrote the contents of Results section based on new results of statistical analysis as below.
“The results of multiple regression analysis examining the factors related to self-reported depressive symptoms are shown in Table 2. Model 1 demonstrated the association of covariates, including gender, age, ADHD symptoms and perceived family function with self-reported depressive symptoms (F (df) = 5.101 (4, 190), p = .001, change of R2 = .097). Model 2 included experiences of bullying involvement in addition to covariates (F (df) = 6.372 (8, 186), p < .001, change of R2 = .118). The results of Model 2 demonstrated that after controlling for the effects of covariates, perpetrators of traditional bullying had more severe self-reported depressive symptoms than did nonperpetrators. Model 3 included frustration discomfort and hostility in addition to covariates and experiences of bullying involvement (F (df) = 10.772 (10, 184), p < .001, change of R2 = .154). The results of Model 3 showed that frustration intolerance was positively associated with self-reported depressive symptoms.” Please refer to line 303-314.
“The results of logistic regression analysis examining the factors related to suicidality are shown in Table 3. Model 1 demonstrated the association of covariates with suicidality. Model 2 included experiences of bullying involvement in addition to covariates. The results of Model 2 demonstrated that after controlling for the effects of covariates, cyberbullying victims were more likely to have suicidality than nonvictims. Model 3 included frustration discomfort and hostility in addition to covariates and experiences of bullying involvement. The results of Model 3 showed that hostility was positively associated with suicidality.” Please refer to line 317-324.
Comment 3
Please specify the softwares/packages used for the statistical analysis
Response
We added it into “2.3. Statistical Analysis”. Please refer to line 277-278.
“We performed data analysis using IBM SPSS Statistics version 24.0 software (IBM Corp., Armonk, NY, USA).
Comment 4
In the abstract, the phrase "multiple and logistic regression analyses" is used. The analysis methods adopted are not clear: logistic regressions are also multiple regressions in this case. I suggest simply using "multiple regressions".
Response
Thank you for your suggestion. We simplified the phrase "multiple and logistic regression analyses" into "multiple regressions". Please refer to line 26.
Comment 5
I suggest avoiding the phrasing "significant variables": it's not variables that can be significant, rather the relationships between variables
Response
Thank you for your suggestion. We deleted the phrase "significant variables" from the revised manuscript. Please refer to line 278, 303 and 317.
Comment 6
I wonder why no normative group was included in the study. It would have been useful to check whether the association between bullying-related variables and suicidality and depression are different (e.g. stronger) in ADHD individuals. In particular, authors seem interested in the comparison between ADHD and non-ADHD adolescents (see e.g. paragraph 4.1), so this would be very useful to test their hypotheses. I think this should be at least included as a limitation.
Response
Thank you for your comment. We listed it as one of limitations in this study as below. Please refer to line 409-413.
“Moreover, this study did not include adolescents without ADHD; therefore, we could not determine whether the associations of bullying involvement, frustration intolerance, and hostility with depressive symptoms and suicidality found in this study also exists in adolescents without ADHD.”
Comment 7
Scoring is not clear for some variables: e.g. Frustration intolerance: were means/medians used? The rationale for coding some variables binomially should also be specified.
Response
- We revised the paragraph describing the variables expressed as means, SD, and percentages as below.
“Adolescents’ gender, experiences of bullying involvement, and suicidality were expressed as percentages. Adolescents’ age, ADHD symptoms, perceived family function, frustration intolerance, hostility, and self-reported depressive symptoms were expressed as means and standard deviations (SDs).” Please refer to line 278-281.
- We also explained that the variables of bullying involvement were coded binomially “because of the skewness of the data.” Please refer to line 213 and 230.
Reviewer 3 Report
The manuscript by Liu et al. entitled “Depression and Suicidality in Adolescents with Attention-Deficit/Hyperactivity Disorder: Roles of Bullying Involvement, Frustration Intolerance, and Hostility” investigates the association between cyberbullying and traditional bullying victimization and perpetration, perceived family function, frustration discomfort, and hostility with self-reported depressive signs, and suicidality in a sample of 195 Taiwanese individuals aged between 11 and 18.
The topic addressed in the manuscript is very interesting; evidence about the relationships between the aforementioned variables is controversial, however, I think that some critical points must be tackled.
Major concern
The statistical analyses that the authors performed are not consistent with the analysis of the literature and the aims of the study. For instance, the authors do not mention the impact of gender and bullying on self-reported depressive signs among the adolescent in Taiwan but the authors conducted some t-test to examine this. Overall, the analysis of the literature is the theoretical framework, the starting point to implement your study (e.g., aims, hypotheses, choice of the statistical analyses). The feeling is that there is not consistency between the state of the art, aims and statistical analyses that you conduct.
In the paper the authors argue about depression but in the material section they mention only the administration of the CES-D that is a self-report questionnaire to detect the occurrence of depressive symptoms not to diagnose depression. Therefore, if the diagnosis of depression was entirely based on the administration of the CES-D, this a very serious methodological bias, since you risk overestimating the occurrence of a clinical condition that is not actually present in your sample. You can talk about depression only if the diagnosis is conducted by an expert (e.g., child psychiatrists that you mentioned in the Participants section) conducting interview, using different resources (e.g., proxies, DSM criteria) and not just a self-report inventory. It is more appropriate to use expression such as ‘self-reported depressive signs/symptoms’ if what you called ‘depression’ is based only in the administration of the CES-D as I suppose after reading the paper.
The results section must be carefully revised and after that all the results must be discussed in the discussion section, establishing a link between ALL your findings and previous evidence. A series of Tables are presented in the Results section, some of them are incomplete (e.g., the table reporting the correlational analyses cannot be interpreted as it is not clarified the association of what variables has been examined), and in the main text a clear description of the analyses performed is missing. Overall, I think that this section is confusing and inconsistent with the declared aim of the study. Finally, the authors illustrated some t-test analyses to examine the impact of gender on some measures (e.g., self-reported depressive symptoms). However, this analysis cannot be performed if an independent variable such as gender is not equally distributed across your participants. This is a very critical issue, since the risk of making a Type 1 error (i.e., we reject the null hypothesis, when it is, in fact, true) is evident.
Page 1 in the abstract: the authors stated that in the regression analyses they controlled the effect of ‘other factors’. What factors? A clarification here is necessary because reading the Results section, it does not seem to me that some factors was controlled for in the logistic regression (dependent variable: Suicidality).
Page 1, in the abstract the authors firstly stated that they performed some regression analyses and they they report the significant associations that they found. In my opinion, it would be more correct that first the authors report the results of some correlational analyses, and they clarify which factors predict the examined dependent variables. The fact that there are some associations between some factors does not necessarily imply tha some of those factors may be significant predictors.
Page 1-2: paragraph 1.1. I think that a more detailed description about ADHD which is the focus of the paper would be helpful to the less expert readers. For instance, at page 1 the authors state “Depression and suicidality are prevalent in individuals with ADHD”. My question is: how much? Please provide some epidemiological evidence, if it is possible concerning the occurrence of the aforementioned neurodevelopmental disorder in the young population in Taiwan, its onset, the clinical subtypes and the occurrence of depression and suicidality in the Taiwanese adolescents with ADHD, if this information is available.
Page 2: I think that paragraph 1.2 must be developed. At present the authors present only a list of possible factors associated with depression and suicidality in teenagers with ADHD. The analysis of the literature, the so called ‘state of the art’ which is the theoretical basis of your study needs to be carefully presented.
Page 2. In the paragraph 1.3 the authors illustrate the association between traditional bullying victimization and negative mood and suicidality. However, at present an operational definition of what ‘traditional bullying victimization’ and cyberbullying is lacking. Please, provide adequate operational definitions of both these constructs.
Page 2. Paragraph 1.4. Can you provide an overt operational definition of ‘irrational belief’? Some examples could be also very helpful.
Page 3: Study aims. The authors posit that they examined the associations between a series of variables. It seems to me that the goal is inconsistent with the analyses described in the Results section. Indeed the regression analyses do not examine if there are significant relationships (in this case you just perform some correlational analyses) rather it seems to me that you explored whether some factors predict depressive signs and suicidality in adolescents with ADHD experiencing victimization and different forms of bullying. Moreover, the analyses that you report are inconsistent with this aim. For instance, here you don’t mention that you examined the impact of gender and different types of bullying on self-reported CES-D. This section to me must be revised so that the aims are consistent with the statistical analyses. Moreover, after that you illustrate the aims, you must present your hypotheses.
Page 3: section 2.2.1. Was the internal consistency calculated in your sample? Did you report the alpha reported in the original validation of the tool in the adolescents of Taiwan? Please clarify and following APA guide, report the Cronbach’s alpha of your sample for each tool that you administered.
Page 3: section 2.2.1 As the authors correctly report, the CES-D is a tool to self-rate the occurrence of depressive symptoms. Therefore, instead of writing ‘Depression’ in the title of this section, I suggest to write ‘Self-reported depressive symptoms’, because you can’t diagnose depression basing your assessment only on the administration of a self-report inventory.
Page 3: section 2.2.1. The authors state “A higher total score indicates more severe depression”. To me this is not correct for two reason. First, as I reported earlier, the CES-D is a self-report measure of the occurrence of depression signs and not a tool to diagnose depression. So you can refer to depressive signs, depressive symptoms but it is incorrect to talk about depression. You can talk about suspected depression. Second, from a psychometrical perspective, as I reported also earlier, higher total score does not mean anything if you do not report what the cut-off that among the adolescents in Taiwan is to argue that an individuals exhibit significant or not significant depressive signs.
Page 4: section 2.2.5, section 2.2.6, and section 2.2.7. The authors state “A higher total score”. Again, higher implies a comparison (i.e., higher than…?). Please clarify what you mean here, providing clear information about the cut-off scores that you used in all the self-report measures that you used in your study.
Page 5 section 2.3. What do the t-tests add to the regression analyses? Conducting both the t-test and examining after them if gender for instance is a predictor of self-reported depressive signs is redundant. In my opinion the regression analyses are sufficient.
Page 5. Results section. The authors should also conduct the non parametric tests (i.e., chi squared) to test whether the occurrence of victims and perpetrator in both traditional bullying and cyberbullying conditions were equally distributed. Moreover, in the main text, the analyses have to be reported including degree of freedom (dfs), p, effect size (when requested) and then you can summarize your findings in some tables. Moreover, the authors do not explain why in the logistic regression the sociodemographic variables (i.e., gender, age) where not included. Controlling for the effect of them for instance when you perform the logistic regression on the Suicidality score is more appropriate and correct than conducting the t-test to examine the effect of gender on the suicidality score, since in your study it is evident that gender is not equally distributed across the participants and therefore conducting a t-test to examine the effect of gender is wrong from a statistical viewpoint. In the main text of this section the outcomes must be presented reporting all the necessary information. See for instance APA manual.
Reference:
American Psychological Association. (2019). Publication Manual of the American Psychological Association (7th edition). American Psychological Association.
Page 5. Table 1. Despite the lack of the non-parametric tests to examine whether the variable ‘gender’ was equally distributed, the rough data suggest that only 31 females/195 participants took part in the study. How did you perform the t-test using gender like independent variable if it is not equally distributed? What sort of t-test did you apply? If you performed the traditional t-test, from a statistical viewpoint what you did is not correct, since as I reported earlier, in your study gender was not counterbalanced across your participants. In my opinion t-test are not necessary but redundant considering that you also performed the analyses, however, in case a t-test is what you want to conduct, in this case you can only carry out Welch’s test.
Page 6, Table 2: the results of the correlational analyses are not clear. You repoirt the r values but you do not clarify what variables is associated with another variable (e.g., variable 1 and variable 2, variable 1 and variable 3). See for instance APA style (7th edition) to understand how you must report the outcomes of the correlational analyses.
Page 6: The way in which the regression analyses are presented is confusing. Please clarify what the predictors, covariate and dependent variable of each regression analyses are. To correctly report the outcomes of this type of analysis, see for instance the APA manual, where some examples are provided. Otherwise, see for instance Pallant (2016).
Reference
Pallant, G. (2016). SPSS Survival Manual (6th edition). Allen & Unwin.
Page 7 Table 4: chi squared and t-test must be reported separately, you can’t write Chi o t. Moreover, dfs must be indicated. Concerning the analyses that you report, I am wondering whether what you report concerning gender is t-test or chi-squared. In case you conducted a chi square analysis, I do not believe that male and female participants with and without ideation or attempts of suicide are equally distributed. Therefore, even a t-test cannot be conducted using the frequency reported in the table.
Discussion
A clear discussion about the nature of the associations between the variable that you examined is lacking, as well as in the Results section a clear illustration in the main text of the outcomes of the analyses that you conducted is missing. For instance, I still don’t understand what the r values are between the dependent variables that you compared by a series of Pearson’s correlations, if these associations are positive or negative, the entity of the effect sizes and if following Cohen (1988) this correlations were low, moderate or high. Moreover, I did not understand how much variance relative to the Suicidality score was explained by the logistic regression analyses that you performed. Therefore, considering how you presented your findings and how you did not put your results in relationship with the existing literature, I cannot judge the correctness of your discussion.
Minor points:
Page 3. The authors state “Research has shown that hostility is associated with a higher risk of depression”. Higher implies a comparison. Please, clarify what you mean for ‘higher…than?’ compared to what?
Page 3: “higher frustration intolerance” than ??? what is the range to establish normal values and high? Please, clarify
Page 3: in the Participants section the authors illustrated the exclusion criteria. Please, clarify how many participants we excluded.
Page 3: section 2.2.2. is there a cut-off to be used when you administer the Suicidality tool?
Page 4: section 2.2.3. The authors argue that “Participants who scored between 1 and 3 in any of the first three or final three items were identified as cyberbullying victims or perpetrators, respectively”. Do you mean that 3/9 is the total score used to be classified as ‘victim’ or ‘perpetrator’? Please clarify what the cut-off is.
Page 4: section 2.2.4. What is the cut-off that you sed? Please, provide this information.
Page 5, section 2.3. Was the occurrence of depression diagnosed by the child psychiatrists that you mentioned in the Participants section? Because if the occurrence of depression is based only on the administration of the CES-D, this is a serious mistake. You could diagnose depression only if the criteria expressed for instance in the DSM 5 (APA; 2013) are satisfied not just using a self-report inventory. Please, provide adequate information about this very critical issue.
Author Response
We appreciated your valuable comments. As discussed below, we have revised our manuscript with underlines based on your suggestions. Please let us know if we need to provide anything else regarding this revision.
Comment 1
The statistical analyses that the authors performed are not consistent with the analysis of the literature and the aims of the study. For instance, the authors do not mention the impact of gender and bullying on self-reported depressive signs among the adolescent in Taiwan but the authors conducted some t-test to examine this. Overall, the analysis of the literature is the theoretical framework, the starting point to implement your study (e.g., aims, hypotheses, choice of the statistical analyses). The feeling is that there is not consistency between the state of the art, aims and statistical analyses that you conduct.
Response
- Thank you for your comment. In the revised manuscript, we reorganized the contents of Methods section by clarifying what the dependent variables (self-reported depressive symptoms and suicidality), independent variables (victimization and perpetration of cyberbullying and traditional bullying, frustration intolerance, and hostility), and covariates (gender, age, ADHD symptoms, and perceived family function). Please refer to line 179-275. We also revise the content of Study Aims and Statistical Analysis section as below to make the statistical strategies clearer. Moreover, the covariates were labelled in Table 1 and Results section (Tables 2 and 3).
1.5. Study Aims
“This study examined the associations of cyberbullying and traditional bullying victimization and perpetration, frustration discomfort, and hostility (independent variables) with self-reported depressive symptoms and suicidality (dependent variables) in adolescents with ADHD by controlling for the effects of gender, age, ADHD symptoms, and perceived family function (covariates).” Please refer to line 149-153.
2.3. Statistical Analysis
“The associations of experiences of bullying involvement, frustration intolerance, and hostility (independent variables) with self-reported depressive symptoms and suicidality (dependent variables) were examined using multiple regressions by using gender, age, ADHD symptoms, and perceived family function as covariates.” Please refer to line 281-285.
- We added the results of previous studies on the impact of bullying involvement on depressive symptoms in adolescents, as well as the gender effect on depressive symptoms and suicidality in adolescents with ADHD as below.
“A previous study on 6,406 adolescents in Taiwan also confirmed the positive association between victimization and perpetration of traditional bullying with self-reported depressive symptoms [29].” Please refer to line 107-109.
“Research has revealed that female sex [3]… can predict depression-related problems in individuals with ADHD.” Please refer to line 80.
“Studies have also shown that female sex [2,3]… can predict suicidal behaviors.” Please refer to line 83.
- Based on your suggestion, we deleted the preliminary t-tests and Pearson’s correlation. Instead, we used a multiple regression for each dependent variable and obtained the similar results. We replaced the original tables 2 to 5 by tables 2 and 3 presenting the results of multiple regressions for depressive symptoms and suicidality, respectively. We also rewrote the contents of Results section based on new results of statistical analysis as below.
“The results of multiple regression analysis examining the factors related to self-reported depressive symptoms are shown in Table 2. Model 1 demonstrated the association of covariates, including gender, age, ADHD symptoms and perceived family function with self-reported depressive symptoms (F (df) = 5.101 (4, 190), p = .001, change of R2 = .097). Model 2 included experiences of bullying involvement in addition to covariates (F (df) = 6.372 (8, 186), p < .001, change of R2 = .118). The results of Model 2 demonstrated that after controlling for the effects of covariates, perpetrators of traditional bullying had more severe self-reported depressive symptoms than did nonperpetrators. Model 3 included frustration discomfort and hostility in addition to covariates and experiences of bullying involvement (F (df) = 10.772 (10, 184), p < .001, change of R2 = .154). The results of Model 3 showed that frustration intolerance was positively associated with self-reported depressive symptoms.” Please refer to line 303-314.
“The results of logistic regression analysis examining the factors related to suicidality are shown in Table 3. Model 1 demonstrated the association of covariates with suicidality. Model 2 included experiences of bullying involvement in addition to covariates. The results of Model 2 demonstrated that after controlling for the effects of covariates, cyberbullying victims were more likely to have suicidality than nonvictims. Model 3 included frustration discomfort and hostility in addition to covariates and experiences of bullying involvement. The results of Model 3 showed that hostility was positively associated with suicidality.” Please refer to line 317-324.
Comment 2
In the paper the authors argue about depression but in the material section they mention only the administration of the CES-D that is a self-report questionnaire to detect the occurrence of depressive symptoms not to diagnose depression. Therefore, if the diagnosis of depression was entirely based on the administration of the CES-D, this a very serious methodological bias, since you risk overestimating the occurrence of a clinical condition that is not actually present in your sample. You can talk about depression only if the diagnosis is conducted by an expert (e.g., child psychiatrists that you mentioned in the Participants section) conducting interview, using different resources (e.g., proxies, DSM criteria) and not just a self-report inventory. It is more appropriate to use expression such as ‘self-reported depressive signs/symptoms’ if what you called ‘depression’ is based only in the administration of the CES-D as I suppose after reading the paper.
Response
We agreed that what we measured in the present study was self-reported depressive symptoms but not the diagnosis of depressive disorders. We replaced “depression” by “self-reported depressive symptoms” thorough the revised manuscript, including the title. Please refer to the title (line 2) and thorough the manuscript.
Comment 3
The results section must be carefully revised and after that all the results must be discussed in the discussion section, establishing a link between ALL your findings and previous evidence. A series of Tables are presented in the Results section, some of them are incomplete (e.g., the table reporting the correlational analyses cannot be interpreted as it is not clarified the association of what variables has been examined), and in the main text a clear description of the analyses performed is missing. Overall, I think that this section is confusing and inconsistent with the declared aim of the study. Finally, the authors illustrated some t-test analyses to examine the impact of gender on some measures (e.g., self-reported depressive symptoms). However, this analysis cannot be performed if an independent variable such as gender is not equally distributed across your participants. This is a very critical issue, since the risk of making a Type 1 error (i.e., we reject the null hypothesis, when it is, in fact, true) is evident.
Response
Thank you for your suggestion. As described in the response to Comment 1, we deleted the preliminary t-tests and Pearson’s correlation. Instead, we used a multiple regression for each dependent variable and obtained the similar results. We replaced the original tables 2 to 5 by tables 2 and 3 presenting the results of multiple regressions for depressive symptoms and suicidality, respectively. We also rewrote the contents of Results section based on new results of statistical analysis.
Comment 4
Page 1 in the abstract: the authors stated that in the regression analyses they controlled the effect of ‘other factors’. What factors? A clarification here is necessary because reading the Results section, it does not seem to me that some factors was controlled for in the logistic regression (dependent variable: Suicidality).
Response
Thank you for your suggestion. We revised Abstract section and described what we controlled in multiple regression as below. Moreover, the covariates were labelled in Methods section and Tables 1 to 3.
“The results showed that after the effects of gender, age, ADJD symptoms, and family function were controlled,…” Please refer to line 28.
Comment 5
Page 1, in the abstract the authors firstly stated that they performed some regression analyses and they report the significant associations that they found. In my opinion, it would be more correct that first the authors report the results of some correlational analyses, and they clarify which factors predict the examined dependent variables. The fact that there are some associations between some factors does not necessarily imply that some of those factors may be significant predictors.
Response
Thank you for your suggestion. As described in the response to Comment 1, we deleted the preliminary t-tests and Pearson’s correlation. Instead, we used a multiple regression for each dependent variable and obtained the similar results. Therefore, we described the results of multiple regression in Abstract section only. Please refer to line 28-31.
Comment 6
Page 1-2: paragraph 1.1. I think that a more detailed description about ADHD which is the focus of the paper would be helpful to the less expert readers. For instance, at page 1 the authors state “Depression and suicidality are prevalent in individuals with ADHD”. My question is: how much? Please provide some epidemiological evidence, if it is possible concerning the occurrence of the aforementioned neurodevelopmental disorder in the young population in Taiwan, its onset, the clinical subtypes and the occurrence of depression and suicidality in the Taiwanese adolescents with ADHD, if this information is available.
Response
Thank you for your suggestion. In the revised manuscript:
- We added more descriptions for the risks of depressive disorders and suicidality among adolescents with ADHD as below.
“… girls with ADHD had a 2.5 times higher risk for major depressive disorder in adolescence and young adulthood compared with those without ADHD [2]. Children with ADHD at 4 to 6 years of age were at greatly increased risk for major depression or dysthymia (hazard ratio, 4.32) through the age of 18 years relative to comparison children [3]. A US nationally representative household survey reported that adults with ADHD were five times more likely to develop a mood disorder than those without ADHD [4]. The 2007 National Survey of Children’s Health in the United States found that children and adolescents with ADHD were more likely to have depression than those without ADHD (14% vs. 1%) [5].…. children with ADHD at 4 to 6 years of age were at greatly increased risk for attempting suicide (hazard ratio, 3.60) through the age of 18 years relative to comparison children [3].” Please refer to line 48-56 and 59-61.
- We also added the results of studies regarding ADHD in children and adolescents with ADHD and comorbid depressive symptoms/disorders and suicidality in Taiwan as below. Please refer to line 66-76.
“The lifetime prevalence of ADHD among children in Taiwan is 10.1% according to the Diagnostic and Statistical Manual of Mental Disorders, Fifth Edition (DSM-5) diagnostic criteria [9] in a nationally representative sample of children in Taiwan [10]. A nationwide population-based study in Taiwan demonstrated that the individuals with showed a significantly increased probability of developing a depressive disorder when compared to the control group (ADHD: 5.3% vs. control: 0.7%) [11]. Another nationwide population-based study in Taiwan also showed that individuals with ADHD showed the higher mortality caused by suicide (adjusted hazard ratio, 2.09; 95% confidence interval [CI], 1.62-2.71) than those without ADHD [12]. The results of previous studies supported that depression-related problems and suicidality warrants carefully evaluation in the individuals with ADHD.”
Comment 7
Page 2: I think that paragraph 1.2 must be developed. At present the authors present only a list of possible factors associated with depression and suicidality in teenagers with ADHD. The analysis of the literature, the so called ‘state of the art’ which is the theoretical basis of your study needs to be carefully presented. Please refer to line 87-99.
Response
Thank you for your suggestion We added a paragraph as below to illustrate the individual and environmental factors that contribute to depressive symptoms and suicidality in adolescents with ADHD based on Ecological Systems Theory; the paragraph also connects to experiences of bullying involvement, frustration intolerance, and hostility that the present study examined.
“According to Ecological Systems Theory [25], children develop within a complex system of relationships within various levels of their environment. The results of previous study indicated that depressive symptoms and suicidality arise from interactions between children with ADHD and their social environments. Both individual (e.g., gender, age, ADHD symptoms, self-competency, and information-processing biases) and environmental factors (e.g., maternal depression, parent–child difficulties, and problems with peer) contributed to depressive symptoms and suicidality in individuals with ADHD. Therefore, depressive symptoms and suicidality in individuals with ADHD should be considered for prevention and intervention in an ecological view. In addition to the individual and environmental factors that have been examined in previous studies, the roles of victimization and perpetration of cyberbullying and traditional bullying, frustration intolerance, and hostility in depressive symptoms and suicidality in adolescents with ADHD warrants further study.”
Comment 8
Page 2. In the paragraph 1.3 the authors illustrate the association between traditional bullying victimization and negative mood and suicidality. However, at present an operational definition of what ‘traditional bullying victimization’ and cyberbullying is lacking. Please, provide adequate operational definitions of both these constructs.
Response
Thank you for your comment. We added the operational definitions of traditional bullying and cyberbullying as below into the revised manuscript.
“Bullying is the activity of repeated, aggressive behavior intended to hurt another individual, physically, mentally, or emotionally [26]. Traditional bullying can involve physical acts, verbal utterances, social exclusion, property theft, and other behaviors [27].” Please refer to line 102-104.
“Cyberbullying refers to the bullying behaviors perpetrated through electronic means [30].” Please refer to line 109-110.
Comment 9
Page 2. Paragraph 1.4. Can you provide an overt operational definition of ‘irrational belief’? Some examples could be also very helpful.
Response
Thank you for your suggestion. We added an example of irrational belief commonly seen in people with high frustration intolerance as below into the revised manuscript.
“…people with high frustration intolerance may strongly demand that existing conditions must be changed to give them what they like, otherwise they can't stand it at all [41].” Please refer to line 137-139.
Comment 10
Page 3: Study aims. The authors posit that they examined the associations between a series of variables. It seems to me that the goal is inconsistent with the analyses described in the Results section. Indeed the regression analyses do not examine if there are significant relationships (in this case you just perform some correlational analyses) rather it seems to me that you explored whether some factors predict depressive signs and suicidality in adolescents with ADHD experiencing victimization and different forms of bullying. Moreover, the analyses that you report are inconsistent with this aim. For instance, here you don’t mention that you examined the impact of gender and different types of bullying on self-reported CES-D. This section to me must be revised so that the aims are consistent with the statistical analyses. Moreover, after that you illustrate the aims, you must present your hypotheses.
Response
Thank you for your comment. We revised the contents of “1.5. Study Aims” and “2.3. Statistical Analysis” to illustrate more clear our study aims, hypotheses, and correspondent statistical strategies as below. Dependent variables, independent variables, and covariates (including gender) were further described to make the statistical strategies more clearer.
1.5. Study Aims
“This study examined the associations of cyberbullying and traditional bullying victimization and perpetration, frustration discomfort, and hostility (independent variables) with self-reported depressive symptoms and suicidality (dependent variables) in adolescents with ADHD by controlling for the effects of gender, age, ADHD symptoms, and perceived family function (covariates). Accordingly, the specific hypotheses are as follows:
Hypothesis 1 (H1). Adolescents with ADHD who are victims or perpetrators of cyberbullying and traditional bullying had severer self-reported depressive symptoms and were more likely to report suicidality than nonvictims and nonperpetrators.
Hypothesis 2 (H2). Greater frustration discomfort and hostility were significantly associated with self-reported depressive symptoms and suicidality in adolescents with ADHD.” Please refer to line 149-158.
2.3. Statistical Analysis
“The associations of experiences of bullying involvement, frustration intolerance, and hostility (independent variables) with self-reported depressive symptoms and suicidality (dependent variables) were examined using multiple regressions by using gender, age, ADHD symptoms, and perceived family function as covariates.” Please refer to line 281-285.
Comment 11
Page 3: section 2.2.1. Was the internal consistency calculated in your sample? Did you report the alpha reported in the original validation of the tool in the adolescents of Taiwan? Please clarify and following APA guide, report the Cronbach’s alpha of your sample for each tool that you administered.
Response
Thank you for your comment. We added the original validation and reliability of the tools in Taiwanese population and the Cronbach’s alpha of the current sample as below.
CES-D
“The Mandarin Chinese version of the CES-D has good validity in discriminating both major depressive disorder (area under the ROC curves [AUCs] = 0.90) and dysthymic disorder (AUC = 0.94), excellent internal consistency (α = .93), and good test–retest reliability (r = .78) [43,44]. In the current sample, internal consistency was good (α = .84).” Please refer to line 189-195.
Suicidality
“A previous study on Taiwanese adolescents reported that its validity was acceptable (Kappa coefficient of agreement between adolescents’ self-report and their parents' reports: .541; p <0.001) [58].” Please refer to line 202-203.
Cyberbullying Experience Questionnaire
“A previous study on adolescents with ADHD using the cutoff of the CEQ demonstrated the significant association between cyberbullying victimization and traditional bullying victimization as well as the associations of cyberbullying perpetration with traditional bullying perpetration and internet addiction (α = .70 for victimization and .64 for perpetration) [38]. In the current sample, internal consistency was .70 and .65 for cyberbullying victimization and perpetration, respectively.” Please refer to line 215-220.
School Bullying Experience Questionnaire
“The C-SBEQ has acceptable validity in discriminating victims and perpetration (Kappa coefficient of agreement between adolescents’ self-report and their teachers’ and classmates’ nomination: .52 for victims and 0.45 for perpetrators; p < .001), acceptable internal consistency (α = .73 for victimization and .76 for perpetration), and acceptable test–retest reliability (r = .80 for victimization and .76 for perpetration) among adolescents in Taiwan [59]. In the current sample, internal consistency was acceptable (α = .76 for victimization and .72 for perpetration).” Please refer to line 232-238.
Frustration Intolerance
“A total score indicates the level of frustration intolerance. The Mandarin Chinese version of the FDS has excellent internal consistency among adolescents with in Taiwan (α = .90) [62]. In the current sample, internal consistency was good (α = .88).” Please refer to line 244-247.
Hostility
“A total score represents the level of hostility. The BDHIC-SF has excellent internal consistency (α = .93) and good test–retest reliability (r = .80) in Taiwanese population [64]. In the current sample, internal consistency was good (α = .85).” Please refer to line 253-256.
Chinese version of the Child Behavior Checklist (CBCL) For Ages 6–18
“The CBCL For Ages 6–18 has good internal consistency (α = .82) in Taiwanese children and adolescents [67]. In the current sample, internal consistency was good (α = .81).” Please refer to line 265-267.
Family APGAR Index
“The Mandarin Chinese version of the Family APGAR Index has excellent internal consistency (α = .88) in Taiwanese adolescents [68]. In the current sample, internal consistency was good (α = .81).” Please refer to line 272-275.
Comment 12
Page 3: section 2.2.1 As the authors correctly report, the CES-D is a tool to self-rate the occurrence of depressive symptoms. Therefore, instead of writing ‘Depression’ in the title of this section, I suggest to write ‘Self-reported depressive symptoms’, because you can’t diagnose depression basing your assessment only on the administration of a self-report inventory.
Response
We agreed that what the CES-D measured was self-reported depressive symptoms. We replaced “depression” by “self-reported depressive symptoms” thorough the revised manuscript, including the title. Please refer to the title (line 2) and thorough the manuscript.
Comment 13
Page 3: section 2.2.1. The authors state “A higher total score indicates more severe depression”. To me this is not correct for two reason. First, as I reported earlier, the CES-D is a self-report measure of the occurrence of depression signs and not a tool to diagnose depression. So you can refer to depressive signs, depressive symptoms but it is incorrect to talk about depression. You can talk about suspected depression. Second, from a psychometrical perspective, as I reported also earlier, higher total score does not mean anything if you do not report what the cut-off that among the adolescents in Taiwan is to argue that an individuals exhibit significant or not significant depressive signs.
Response
Thank you for your reminding. We replaced “depression” by “self-reported depressive symptoms” thorough the revised manuscript. Moreover, we revised the statement “A higher total score indicates more severe depression” into “A total score indicates the severity of self-reported depressive symptoms.” Please refer to line 189-190.
Comment 14
Page 4: section 2.2.5, section 2.2.6, and section 2.2.7. The authors state “A higher total score”. Again, higher implies a comparison (i.e., higher than…?). Please clarify what you mean here, providing clear information about the cut-off scores that you used in all the self-report measures that you used in your study.
Response
Thank you for your comment. We used the total scores but not cutoff to represent the levels of perceived family function, frustration intolerance, and hostility. Therefore, we revised the statements “A higher total score…” into the new statements as below.
Perceived Family Function
“A total score represents the level of family support.” Please refer to line 272-273.
Frustration Intolerance
“A total score indicates the level of frustration intolerance.” Please refer to line 244-245.
Hostility
“A total score represents the level of hostility. ” Please refer to line 253.
Comment 15
Page 5 section 2.3. What do the t-tests add to the regression analyses? Conducting both the t-test and examining after them if gender for instance is a predictor of self-reported depressive signs is redundant. In my opinion the regression analyses are sufficient.
Response
Thank you for your suggestion. As stated in Response to Comment 1-3, we deleted the preliminary t-tests and Pearson’s correlation. Instead, we used a multiple regression for each dependent variable and obtained the similar results. We replaced the original tables 2 to 5 by tables 2 and 3 presenting the results of multiple regressions for depressive symptoms and suicidality, respectively. We also rewrote the contents of Results section based on new results of statistical analysis.
Comment 16
Page 5. Results section.
- The authors should also conduct the non parametric tests (i.e., chi squared) to test whether the occurrence of victims and perpetrator in both traditional bullying and cyberbullying conditions were equally distributed.
- Moreover, in the main text, the analyses have to be reported including degree of freedom (dfs), p, effect size (when requested) and then you can summarize your findings in some tables.
- Moreover, the authors do not explain why in the logistic regression the sociodemographic variables (i.e., gender, age) where not included. Controlling for the effect of them for instance when you perform the logistic regression on the Suicidality score is more appropriate and correct than conducting the t-test to examine the effect of gender on the suicidality score, since in your study it is evident that gender is not equally distributed across the participants and therefore conducting a t-test to examine the effect of gender is wrong from a statistical viewpoint. In the main text of this section the outcomes must be presented reporting all the necessary information. See for instance APA manual.
Response
Thank you for your comments. Based on your comments, we made the revisions as below.
- We conducted chi-square test to compare the occurrence of victims and perpetrator of cyberbullying between victims and nonvictims and between perpetrators and nonperpetrators of traditional bullying as below. Please refer to line 294-298.
“Victims of traditional bullying were more likely to be victims of cyberbullying than non-victims of traditional bullying (χ2 = 9.275, p = .002), whereas no difference in the risk of being cyberbullying perpetrators was found between perpetrators and nonperpetrators of traditional bullying (χ2 = 1.346, p = .246).”
- In the revised manuscript we added F value with degree of freedom and p value for multiple regression in the text and Table 2. The information of outcomes was revised into the APA format. We also used R2 (Cohen, 1988) and OR with 95% CI (Cohen and Chen, 2010) to represent effect sizes in multiple regression and logistic regression. Please refer to line 285-288.
“R-squared (R2) was used to represent the effect size of the variables in multiple regression analysis [70]. Odds ratios (ORs) and 95% CIs were used to represent statistical significance and effect sizes of the variables in logistic regression analysis [71].”
- Thank you for your suggestion. As described in the response to Comment 1, we deleted the preliminary t-tests and Pearson’s correlation. Instead, we used a multiple regression for each dependent variable and obtained the similar results.
Comment 17
Page 5. Table 1. Despite the lack of the non-parametric tests to examine whether the variable ‘gender’ was equally distributed, the rough data suggest that only 31 females/195 participants took part in the study. How did you perform the t-test using gender like independent variable if it is not equally distributed? What sort of t-test did you apply? If you performed the traditional t-test, from a statistical viewpoint what you did is not correct, since as I reported earlier, in your study gender was not counterbalanced across your participants. In my opinion t-test are not necessary but redundant considering that you also performed the analyses, however, in case a t-test is what you want to conduct, in this case you can only carry out Welch’s test.
Response
Thank you for your suggestion. As described in the response to Comment 1, we deleted the preliminary t-tests and Pearson’s correlation. Instead, we used a multiple regression for each dependent variable and obtained the similar results.
Comment 18
Page 6, Table 2: the results of the correlational analyses are not clear. You repoirt the r values but you do not clarify what variables is associated with another variable (e.g., variable 1 and variable 2, variable 1 and variable 3). See for instance APA style (7th edition) to understand how you must report the outcomes of the correlational analyses.
Response
Thank you for your suggestion. As described in the response to Comment 1, we deleted the preliminary t-tests and Pearson’s correlation. Instead, we used a multiple regression for each dependent variable and obtained the similar results.
Comment 19
Page 6: The way in which the regression analyses are presented is confusing. Please clarify what the predictors, covariate and dependent variable of each regression analyses are. To correctly report the outcomes of this type of analysis, see for instance the APA manual, where some examples are provided. Otherwise, see for instance Pallant (2016).
Reference
Pallant, G. (2016). SPSS Survival Manual (6th edition). Allen & Unwin.
Response
Thank you for your comment. As stated in the response to Comment 1, we reorganized the contents of Methods section by clarifying what the dependent variables (self-reported depressive symptoms and suicidality), independent variables (victimization and perpetration of cyberbullying and traditional bullying, frustration intolerance, and hostility), and covariates (gender, age, ADHD symptoms, and perceived family function). Please refer to line 179-275. We also revise the content of Study Aims (line 149-153) and Statistical Analysis (line 281-285) section to make the statistical strategies clearer. Moreover, the covariates were labelled in Table 1 and Results section (Tables 2 and 3). We also revised the formats of tables based on APA format.
Comment 20
Page 7 Table 4: chi squared and t-test must be reported separately, you can’t write Chi o t. Moreover, dfs must be indicated. Concerning the analyses that you report, I am wondering whether what you report concerning gender is t-test or chi-squared. In case you conducted a chi square analysis, I do not believe that male and female participants with and without ideation or attempts of suicide are equally distributed. Therefore, even a t-test cannot be conducted using the frequency reported in the table.
Response
Thank you for your suggestion. As described in the response to Comment 1, we deleted the preliminary t-tests and Pearson’s correlation. Instead, we used a multiple regression for each dependent variable and obtained the similar results.
Comment 21
Discussion
- A clear discussion about the nature of the associations between the variable that you examined is lacking, as well as in the Results section a clear illustration in the main text of the outcomes of the analyses that you conducted is missing. For instance, I still don’t understand what the r values are between the dependent variables that you compared by a series of Pearson’s correlations, if these associations are positive or negative, the entity of the effect sizes and if following Cohen (1988) this correlations were low, moderate or high.
- Moreover, I did not understand how much variance relative to the Suicidality score was explained by the logistic regression analyses that you performed. Therefore, considering how you presented your findings and how you did not put your results in relationship with the existing literature, I cannot judge the correctness of your discussion.
Response
Thank you for your suggestion.
- As described in the response to Comment 1, we deleted the preliminary t-tests and Pearson’s correlation. Instead, we used a multiple regression for each dependent variable and obtained the similar results.
- We used OR with 95% CI to represent effect sizes in logistic regression (Cohen and Chen, 2010). Please refer to line 286-288.
“Odds ratios (ORs) and 95% CIs were used to represent statistical significance and effect sizes of the variables in logistic regression analysis [71].”
Comment 22
Minor points:
Page 3. The authors state “Research has shown that hostility is associated with a higher risk of depression”. Higher implies a comparison. Please, clarify what you mean for ‘higher…than?’ compared to what?
Response
Thank you for your comment. We revised this sentence into “Research has shown that hostility is associated with increased risks of depressive disorders [45] and suicidality [46].” Please refer to line 143-144.
Comment 23
Page 3: “higher frustration intolerance” than ??? what is the range to establish normal values and high? Please, clarify
Response
It is the result of comparing the level of frustration intolerance between individuals with and without ADHD. Please refer to line 144-146.
Comment 24
Page 3: in the Participants section the authors illustrated the exclusion criteria. Please, clarify how many participants we excluded.
Response
We added the information of excluded adolescents and caregivers as below. Please refer to line 169-171.
“In total, 216 adolescents with ADHD and their caregivers visited the outpatient clinics during the period of study. Of them, 8 adolescents were excluded based on the exclusion criteria.”
Comment 25
Page 3: section 2.2.2. is there a cut-off to be used when you administer the Suicidality tool?
Response
Thank you for your reminding. We added the sentence as below to represent the cut-off. Please refer to line 204-205.
“Participants responding ‘‘yes’’ to any of the five items were classified as having suicidality. We added it into the revised manuscript.”
Comment 26
Page 4: section 2.2.3. The authors argue that “Participants who scored between 1 and 3 in any of the first three or final three items were identified as cyberbullying victims or perpetrators, respectively”. Do you mean that 3/9 is the total score used to be classified as ‘victim’ or ‘perpetrator’? Please clarify what the cut-off is.
Response
Thank you for your comment. We revised the sentence to clarify the cut-off as below. Please refer to line 213-215.
“Because of the skewness of the data, participants who scored 1 or above to any of the first three or final three items were identified as cyberbullying victims or perpetrators, respectively.”
Comment 27
Page 4: section 2.2.4. What is the cut-off that you sed? Please, provide this information.
Response
We revised the sentence to clarify the cut-off as below. Please refer to line 230-232.
“Because of the skewness of the data, participants who scored 2 or 3 points to any of the first eight or final eight items were identified as victims or perpetrators, respectively, of social, verbal, and physical bullying.”
Comment 28
Page 5, section 2.3. Was the occurrence of depression diagnosed by the child psychiatrists that you mentioned in the Participants section? Because if the occurrence of depression is based only on the administration of the CES-D, this is a serious mistake. You could diagnose depression only if the criteria expressed for instance in the DSM 5 (APA; 2013) are satisfied not just using a self-report inventory. Please, provide adequate information about this very critical issue.
Response
We agreed that what we measured in the present study was self-reported depressive symptoms but not the diagnosis of depressive disorders. We replaced “depression” by “self-reported depressive symptoms” thorough the revised manuscript, including the title. Please refer to the title (line 2) and thorough the manuscript.
Round 2
Reviewer 2 Report
I commend the authors for their thorough efforts to address my original comments. I have no further request and I think this manuscript is now acceptable for publication.
Author Response
Thank you for your support.
Reviewer 3 Report
The manuscript by Liu et al. originally entitled “Depression and Suicidality in Adolescents with Attention-Deficit/Hyperactivity Disorder: Roles of Bullying Involvement, Frustration Intolerance, and Hostility” has been revised, following only partially my requests of clarifications/revision. I appreciate that the authors tried to improve the quality of their article but in my opinion, the weaknesses that the article still presents are too many and I don’t recommend its publication. The reason why I don’t think that the article deserves to be published will be illustrated below.
Even though the authors provided the operational definitions that I requested (e.g., cyberbullying), the analysis of the literature is still very superficial and the ratio underpinning the study is not clear.
As I pointed out in the previous revision, relatively to paragraph 1.1., I suggested presenting a more detailed description of the ADHD disorder, which is the focus of the paper, but at present, it lacks. The authors just illustrated the epidemiological evidence of ADHD in Taiwan, that’s fine but in my opinion, it is not sufficient.
Moreover, in paragraph 1.2, the development of the ‘state of the art’ that I already requested is lacking. Again, the authors present a list of factors associated with the occurrence of ADHD (see page 2, lines 78-86) but they don’t explain appropriately what they are. For instance, writing that ADHD is accompanied by “executive function deficits” (see page 2, line 85) is too generic and superficial, what sort of EFs are not efficient in individuals with ADHD? This is crucial, also to understand for instance the risk that suicidality can be related to increased impulsivity, which, in turn, is strictly related to an inhibitory deficit (i.e., response inhibition is a very crucial EF in the ADHD disorder).
On page 3, in paragraph 1.3 the authors argue about the evidence of a “causal relationship of traditional bullying victimization with depressive symptoms and suicidal ideation and suicide attempts” (lines 104-106). What does it mean? This should be explained properly since it should be the theoretical basis of the study. Specifically, the authors should highlight that bullying victimization is causative of depressive symptoms and further mental problems.
Furthermore, at lines 107-110 the authors argue about a positive relationship “between victimization and perpetration of traditional bullying with self-reported depressive symptoms”. Also, this should be described better.
Overall, at the end of the introduction, it is still missing a clear statement of the issues ‘open’ in the literature driving the study. What is the gap that the authors want to overcome?
Relatively to the aim of the study (see page 4, lines 149-152), again, as I already reported in my previous review, the goal of the study does not fit with the analyses that the authors reported. If the aim that they declare is to examine a series of relationships, then the regression analyses are not the type of analyses that you conduct. Furthermore, when you examine some relationships (e.g., performing Pearson’s correlations) you don’t have dependent and independent variables BUT you just explore the nature of the relationship between the dependent variables.
The hypotheses are inconsistent with the unique aim that the authors reported. Specifically, relatively to Hypothesis 2, it is not clear if the authors hypothesized a positive or negative relationship between the factors, since they wrote that they expected greater frustration discomfort, but they did not specify anything about the self-reported depressive symptoms (e.g., did they expected more depressive signs in relationship to greater frustration discomfort more? Did they expect less depressive signs?). What is the evidence driving the hypotheses?
Again, the hypotheses are inconsistent with the statistical analyses.
Relatively to the tools administered, I already asked in my previous review to report the cut-off, which is the usual information reported in the literature to understand if for instance the self-reported depressive symptoms assessed through the CES-D measure are significant or their occurrence can be considered within the range of normality. In the current version of the manuscript, the authors did not provide any cut-off index for the tools that they administered. They added some information about how they calculate the score but they did not provide clear information about what can be considered ‘normal’ and what it isn’t. it is fundamental to understand what the cut-off used is and it is essential also reporting the reference of the validation of the tool according to which the cut-off has been extracted.
Moreover, a clear operational definition about the subscales of the Hostility tool (i.e., hostility cognition, hostility affect, expressive hostility behavior, and suppressive hostility behavior) is still lacking. Cut-off, also in this case is lacking.
At the end of the Participants section, a table reporting the socio-demographic characteristics of the participants (age, gender, level of education,) organized as a function of the type of bullying (i.e., traditional vs. cyberbullying) and role (i.e., victim, perpetrator) would be appreciated. Concerning this issue, how did the authors categorize those participants which could not be classified as victims or perpetrators?
In my opinion, apart from the theoretical weakness of this study, a further major problem is represented by the statistical analyses conducted by the authors. I strongly recommend them to be supported by an expert in statistics because what they state is wrong and the analyses that they report to pursue their goal are not correct. Indeed, in the abstract, on page 1, the authors state that “Multiple regression analyses were used to examine the correlates of self-reported depressive symptoms…”. This is a mistake, if you want to explore some relationships, you conduct some correlational analyses since the regression analyses are performed to evaluate the strength of the relationship between the dependent variable and several predictors. Therefore, as I already wrote in the previous review, you can’t perform some regression analyses if before you don’t carry out the correlational analyses between the factors. In other words, the correlational analyses are mandatory to conduct, based on their results, some regression analyses. Furthermore, you need to perform the correlational analyses to exclude multicollinearity between the factors (see Pallant, 2016. The reference was already provided in my previous revision).
A further limit is represented by how the authors report the regression analyses. I already wrote in my previous review how the Results section/tables must be reported (e.g., in APA style). Even though the authors reported some Tables in a better-organized fashion, at present, I cannot assess if these Tables are appropriate, because it is not clear how the regression analyses were conducted (what method?). Indeed, the authors did not provide any information about the method used to perform the regression analyses. In the main text, no information has been provided about what factors were used as predictors and in which order the variables were inserted, it is not clear reading the Results section if the authors used some covariates. I inferred this information from Table 1 (e.g., gender perhaps?) but more detailed information is necessary to be reported in the main text. Furthermore, it should be clearly reported the numerosity of the sample used to perform the regression analyses. Similarly, in the results section and even in Table 2, it is not clear how factors such as “Cyberbullying victims” have been coded to conduct the regression analysis. Moreover, the authors don’t report any information if they conducted some preliminary analyses to ensure no violation of the assumptions of normality, linearity, and homoscedasticity. This is fundamental (considering that I expect that the authors conducted some preliminary data analyses to examine the type of distribution of their data).
Relatively to Table 1, it is not clear how the participants were classified in the category of bullying and cyberbullying. For instance, it is not clear to me if the same participant could be classified twice as a victim in the categories of cyberbullying and traditional bullying or whether one category excluded the other one.
Relatively to the gender factor, in the regression analyses, this factor must be inserted as dichotomous variables, not like the percentage of females or males (as it is reported in Table 1). Perhaps the authors did correctly, but this information has not been reported in the Results section, therefore I cannot assess if the regression analysis is correct or not.
In Table 2, in model 2, gender should be indicated as significant, I believe that the authors did not insert an asterisk (if it is correct that p = .015)
Also how the logistic regression was presented is not correct. Apart from Table3, in the text, a clear description of the factors used, how the dependent variable was coded, are still missing. For instance, how suicidality was assessed? This must be clarified. Furthermore, information reported in Table 3 relatively to the logistic regression is incomplete. Similarly, in the main text, a correct and exhaustive description of the logistic regression is missing (see Pallant, 2016).
The Discussion section reports a misinterpretation of the results. If the analyses that the authors report are correct, the regression analyses pointed out not associations between variables as the authors state, rather the regression analyses tell you which factors predict the dependent variable (if they predict the dv). Moreover, also in the abstract, the authors stated that “greater frustration discomfort and bullying perpetration were significantly associated with more severe self-reported depressive symptoms”. This is wrong since Table 2 shows that only frustration discomfort is a significant predictor of the CES-D score. In my opinion, the Discussion must be written again, after a better understanding of the ratio underlying the analyses that the authors conducted and after conducting the appropriate statistical analyses.
Minor problems:
Abstract, line 28, the authors wrote ‘ADJD’, I guess that they meant ADHD.
The abstract does not provide correct information about the results. How much variance was explained by the regression analyses? What are the significant predictors (this is what you should report)? Again, the regression analyses doesn’t provide information about the relationship but they ‘tell you’ if some factors are predictor of the dv. Therefore, what it is reported also in the abstract is not correct. The statistical limits of your misinterpretation of your results is one of my major concern about the quality of the manuscript, that to me does not reach the minimum standard for its publication.
Author Response
Reviewer 3
Comment 1
Even though the authors provided the operational definitions that I requested (e.g., cyberbullying), the analysis of the literature is still very superficial and the ratio underpinning the study is not clear.
Response
The aims of this study were to examine the associations of cyberbullying and traditional bullying victimization and perpetration, frustration discomfort, and hostility (independent variables) with self-reported depressive symptoms and suicidality (dependent variables) in adolescents with ADHD. We believe that readers can find the importance of examining depressive symptoms and suicidality in adolescents with ADHD in section 1.1 and the roles of these three factors in section 1.2 to 1.4. The operational definition of cyberbullying has been well described in section 1.3 and section 2.2.2.1. Please refer them.
Comment 2
As I pointed out in the previous revision, relatively to paragraph 1.1., I suggested presenting a more detailed description of the ADHD disorder, which is the focus of the paper, but at present, it lacks. The authors just illustrated the epidemiological evidence of ADHD in Taiwan, that’s fine but in my opinion, it is not sufficient.
Response
Firstly, an academic report must focus its contents on the aims of study. As introduced in section 1.1, the present study aimed to examined depressive symptoms and suicidality in adolescents with ADHD; therefore, we presented the important results of previous studies to illustrate the importance of examining depressive symptoms and suicidality in adolescents with ADHD here.
Second, below is your comment in the previous review:
“Page 1-2: paragraph 1.1. I think that a more detailed description about ADHD which is the focus of the paper would be helpful to the less expert readers. For instance, at page 1 the authors state “Depression and suicidality are prevalent in individuals with ADHD”. My question is: how much? Please provide some epidemiological evidence, if it is possible concerning the occurrence of the aforementioned neurodevelopmental disorder in the young population in Taiwan, its onset, the clinical subtypes and the occurrence of depression and suicidality in the Taiwanese adolescents with ADHD, if this information is available.”
So, you can find that we have responded to your comment b y adding the prevalence of depression and suicidality based on the results of previous studies (references 2 to 9). Please refer to section 1.1. We also added the results of previous studies in Taiwan (references 10 to 12) according to you. I am confused that why you said it is not sufficient. In fact, the concept of “subtypes” of ADHD has been abandoned in DSM-5 since 2013. It is out-of-date to mention it as one of rejecting a paper.
Comment 3
Moreover, in paragraph 1.2, the development of the ‘state of the art’ that I already requested is lacking. Again, the authors present a list of factors associated with the occurrence of ADHD (see page 2, lines 78-86) but they don’t explain appropriately what they are. For instance, writing that ADHD is accompanied by “executive function deficits” (see page 2, line 85) is too generic and superficial, what sort of EFs are not efficient in individuals with ADHD? This is crucial, also to understand for instance the risk that suicidality can be related to increased impulsivity, which, in turn, is strictly related to an inhibitory deficit (i.e., response inhibition is a very crucial EF in the ADHD disorder).
Response
As what We have mentioned in the response to Comment 2, an academic report must focus on its main aims; therefore, we used most of the paragraph in Introduction to illustrate depression, suicidality, bullying victimization, frustration intolerance, and hostility. We mentioned that executive function has been identified as a predictor of suicidal behaviors in young people with ADHD in previous studies; therefore, the present study would not examine its prediction. It is obvious that we should not spend a lot of paragraphs to introduce executive function.
Comment 4
On page 3, in paragraph 1.3 the authors argue about the evidence of a “causal relationship of traditional bullying victimization with depressive symptoms and suicidal ideation and suicide attempts” (lines 104-106). What does it mean? This should be explained properly since it should be the theoretical basis of the study. Specifically, the authors should highlight that bullying victimization is causative of depressive symptoms and further mental problems.
Response
- The original sentence in the manuscript is “A meta-analysis has provided strong evidence for a causal relationship of traditional bullying victimization with depressive symptoms and suicidal ideation and suicide attempts [28].” Its meaning is clear.
- Determining the casual relationship between traditional bullying victimization and depressive symptoms is not the purpose of this cross-sectional study. In fact, we listed the cross-sectional study design as one of limitations of this study. The present study examined simultaneously the roles of cyberbullying and traditional bullying involvement in depressive symptoms and suicidality in adolescents with ADHD. This is what the new attempt the present study made.
Comment 5
Furthermore, at lines 107-110 the authors argue about a positive relationship “between victimization and perpetration of traditional bullying with self-reported depressive symptoms”. Also, this should be described better.
Response
The original sentence in the manuscript is “A previous study on 6,406 adolescents in Taiwan also confirmed the positive association between victimization and perpetration of traditional bullying with self-reported depressive symptoms [29].” We believe that this sentence has clearly described the result of reference 29.
Comment 6
Overall, at the end of the introduction, it is still missing a clear statement of the issues ‘open’ in the literature driving the study. What is the gap that the authors want to overcome?
Response
What the present study wanted to overcome the gaps in this topic have been clearly described in the manuscript as below. You can find them easily.
Line 130-132: “...no study has simultaneously examined the roles of cyberbullying and traditional bullying involvement in depressive symptoms and suicidality in adolescents with ADHD.”
Line 146-147: “The relationships of frustration intolerance and hostility with depressive symptoms and suicidality in adolescents with ADHD, however, remain uninvestigated.”
Comment 7
Relatively to the aim of the study (see page 4, lines 149-152), again, as I already reported in my previous review, the goal of the study does not fit with the analyses that the authors reported. If the aim that they declare is to examine a series of relationships, then the regression analyses are not the type of analyses that you conduct. Furthermore, when you examine some relationships (e.g., performing Pearson’s correlations) you don’t have dependent and independent variables BUT you just explore the nature of the relationship between the dependent variables.
Response
It is beyond our understanding why reviewer 3 mentioned about Pearson’s correlation. It has been deleted in the revised manuscript…
Comment 8
The hypotheses are inconsistent with the unique aim that the authors reported. Specifically, relatively to Hypothesis 2, it is not clear if the authors hypothesized a positive or negative relationship between the factors, since they wrote that they expected greater frustration discomfort, but they did not specify anything about the self-reported depressive symptoms (e.g., did they expected more depressive signs in relationship to greater frustration discomfort more? Did they expect less depressive signs?). What is the evidence driving the hypotheses?
Again, the hypotheses are inconsistent with the statistical analyses.
Response
- The hypotheses have been clearly examined using linear regression analysis. Tables 2 and 3 shows the results of multiple regression analysis and logistic regression on the factors related to depressive symptoms and suicidality, respectively. Both of them aimed to examined our hypothesis 1 and hypothesis 2. We believe that readers can figure out them easily.
- We used the whole section 1.4. (line 133 to 147) to illustrate why we hypothesized that greater frustration discomfort and hostility were significantly associated with self-reported depressive symptoms and suicidality in adolescents with ADHD (line 157-158). We believe that readers can fully understand the evidence driving the hypothesis.
Comment 9
Relatively to the tools administered, I already asked in my previous review to report the cut-off, which is the usual information reported in the literature to understand if for instance the self-reported depressive symptoms assessed through the CES-D measure are significant or their occurrence can be considered within the range of normality. In the current version of the manuscript, the authors did not provide any cut-off index for the tools that they administered. They added some information about how they calculate the score but they did not provide clear information about what can be considered ‘normal’ and what it isn’t. it is fundamental to understand what the cut-off used is and it is essential also reporting the reference of the validation of the tool according to which the cut-off has been extracted.
Response
Reviewer 3 totally misunderstood the measures. As we have described in the present study (line 181-182), the CES-D was used to measure the severity of self-reported depressive symptoms. The requirement of “cut-off” for CES-D in this study is unreasonable.
Comment 10
Moreover, a clear operational definition about the subscales of the Hostility tool (i.e., hostility cognition, hostility affect, expressive hostility behavior, and suppressive hostility behavior) is still lacking. Cut-off, also in this case is lacking.
Response
The level of hostility was measured using the Buss–Durkee Hostility Inventory–Chinese Version–Short Form (BDHIC-SF), which assesses participants’ multiple concepts of hostility as described in the manuscript (line 249-251). Again, asking for the “cut-off” of the BDHIC-SF is unreasonable because it violates the aim of this study and the BDHIC-SF.
Comment 11
At the end of the Participants section, a table reporting the socio-demographic characteristics of the participants (age, gender, level of education,) organized as a function of the type of bullying (i.e., traditional vs. cyberbullying) and role (i.e., victim, perpetrator) would be appreciated. Concerning this issue, how did the authors categorize those participants which could not be classified as victims or perpetrators?
Response
- The present study used the victims and perpetrators of traditional bullying and cyberbullying as the independent variables and listed the proportions of them in Table 1. Adding a new table reporting their socio-demographic characteristics will confuse the readers. Therefore, we did not add it.
- The classification of victims and perpetrators of traditional bullying and cyberbullying had been described in the original and revised versions of the manuscript... Please refer to line 213 to 215 and line 230-232.
Comment 12
In my opinion, apart from the theoretical weakness of this study, a further major problem is represented by the statistical analyses conducted by the authors. I strongly recommend them to be supported by an expert in statistics because what they state is wrong and the analyses that they report to pursue their goal are not correct. Indeed, in the abstract, on page 1, the authors state that “Multiple regression analyses were used to examine the correlates of self-reported depressive symptoms…”. This is a mistake, if you want to explore some relationships, you conduct some correlational analyses since the regression analyses are performed to evaluate the strength of the relationship between the dependent variable and several predictors. Therefore, as I already wrote in the previous review, you can’t perform some regression analyses if before you don’t carry out the correlational analyses between the factors. In other words, the correlational analyses are mandatory to conduct, based on their results, some regression analyses. Furthermore, you need to perform the correlational analyses to exclude multicollinearity between the factors (see Pallant, 2016. The reference was already provided in my previous revision).
Response
- In fact, we conducted correlational analysis in the original version of the manuscript. Reviewer 3 might forget it. Because that both Reviewer 1 and Reviewer 2 suggested that we should abandoned correlation analysis and examined directly the relationships between independent and dependent variables by linear regression analysis. Therefore, we deleted the result of correlational analysis.
- Correlational analyses only provide the correlations between variables but did not exclude multicollinearity. We used the conditional index to examine the multicollinearity. Please refer to line 316.
Comment 13
- A further limit is represented by how the authors report the regression analyses. I already wrote in my previous review how the Results section/tables must be reported (e.g., in APA style). Even though the authors reported some Tables in a better-organized fashion, at present, I cannot assess if these Tables are appropriate, because it is not clear how the regression analyses were conducted (what method?). Indeed, the authors did not provide any information about the method used to perform the regression analyses. In the main text, no information has been provided about what factors were used as predictors and in which order the variables were inserted, it is not clear reading the Results section if the authors used some covariates. I inferred this information from Table 1 (e.g., gender perhaps?) but more detailed information is necessary to be reported in the main text.
- Furthermore, it should be clearly reported the numerosity of the sample used to perform the regression analyses. Similarly, in the results section and even in Table 2, it is not clear how factors such as “Cyberbullying victims” have been coded to conduct the regression analysis. Moreover, the authors don’t report any information if they conducted some preliminary analyses to ensure no violation of the assumptions of normality, linearity, and homoscedasticity. This is fundamental (considering that I expect that the authors conducted some preliminary data analyses to examine the type of distribution of their data).
Response
- Table 2 was located next to Table 1. It is impossible for the readers to have no idea what dependent and independent variables are when they read table 1 to table 2. Moreover, the dependent and independent variables have been introduced in Method section, especially in section 2.3. Statistical Analysis. Unless the study used specific methods such as stepwise regression, multiple regression analysis indicates full entered model.
- Reviewer 3 advertised “APA style” but did not mentioned any inadequacy of our presentation. We would like to raise a question: Why did reviewer 3 consider that APA styles should be used for all papers of journals that are not belonged to American Psychological Association (APA)? As we know, most of famous journals around the world do not use APA style.
- We added “(N = 195)” into Table 2. In fact, Table 1 has shown that no participants had missing data, and the data of all participants were used in multiple regression analysis. Independent variables have been introduced in section 1.5. Study Aims and section 2. Methods. We believe that readers understand “Cyberbullying victims” is one of independent variables. We used Shapiro–Wilk test to examine the normality of depressive symptoms, frustration intolerance, and hostility. All p values > .05, indicating they were normally distributed. Please refer to line 294-296. The result of examining multicollinearity was reported in the response to Comment 12.
Comment 14
Relatively to Table 1, it is not clear how the participants were classified in the category of bullying and cyberbullying. For instance, it is not clear to me if the same participant could be classified twice as a victim in the categories of cyberbullying and traditional bullying or whether one category excluded the other one.
Response
Both victims of traditional bullying and victims of cyberbullying are independent variables in the present study. An adolescent could be the victims of traditional bullying and cyberbullying. The similar method of classification could be seen in the studies on adolescent bullying involvement.
Comment 15
Relatively to the gender factor, in the regression analyses, this factor must be inserted as dichotomous variables, not like the percentage of females or males (as it is reported in Table 1). Perhaps the authors did correctly, but this information has not been reported in the Results section, therefore I cannot assess if the regression analysis is correct or not.
Response
What did reviewer 3 mean? We have reported gender as a dichotomous variable in Table 1. We did not mention girls or boys in multiple regression because gender was a covariable here.
Comment 16
In Table 2, in model 2, gender should be indicated as significant, I believe that the authors did not insert an asterisk (if it is correct that p = .015)
Response
Thank you for your comment. We added the asterisk.
Comment 17
Also how the logistic regression was presented is not correct. Apart from Table3, in the text, a clear description of the factors used, how the dependent variable was coded, are still missing. For instance, how suicidality was assessed? This must be clarified. Furthermore, information reported in Table 3 relatively to the logistic regression is incomplete. Similarly, in the main text, a correct and exhaustive description of the logistic regression is missing (see Pallant, 2016).
Response
- We believe that we have reported the results of logistic regression analysis enough to let the readers understand well.
- We do not understand why “how suicidality was assessed” should be repeatedly reported in Results section. It has been introduced in Methods section.
Comment 18
The Discussion section reports a misinterpretation of the results. If the analyses that the authors report are correct, the regression analyses pointed out not associations between variables as the authors state, rather the regression analyses tell you which factors predict the dependent variable (if they predict the dv). Moreover, also in the abstract, the authors stated that “greater frustration discomfort and bullying perpetration were significantly associated with more severe self-reported depressive symptoms”. This is wrong since Table 2 shows that only frustration discomfort is a significant predictor of the CES-D score. In my opinion, the Discussion must be written again, after a better understanding of the ratio underlying the analyses that the authors conducted and after conducting the appropriate statistical analyses.
Response
- We know that regression analysis examined the prediction of independent variables for dependent variable. Based on the principle of correlation studies, “prediction” is an application of correlation. We did not use “prediction” in this cross-section study to avoid the confusion with causal or temporal relationship between independent and dependent variables.
- Reviewer 3 misread Table 2. Both frustration discomfort and perpetrators of traditional bullying were significant predictors of depressive symptoms.
Comment 19
Minor problems:
Abstract, line 28, the authors wrote ‘ADJD’, I guess that they meant ADHD.
Response
We would like to thank reviewer 3 for pointing the typo. We correct it into “ADHD” in line 28.
Comment 20
The abstract does not provide correct information about the results. How much variance was explained by the regression analyses? What are the significant predictors (this is what you should report)? Again, the regression analyses doesn’t provide information about the relationship but they ‘tell you’ if some factors are predictor of the dv. Therefore, what it is reported also in the abstract is not correct. The statistical limits of your misinterpretation of your results is one of my major concern about the quality of the manuscript, that to me does not reach the minimum standard for its publication.
Response
- Variance has been reported in Table 2. We did not think it is necessary to report it in Abstract. The factors significantly associated with depressive symptoms and suicidality have been reported in Abstract. The reason we did not use “predictors” was described in the response to Comment 18.
- We appreciate the comments from Reviewer 1 and Reviewer 2.